# Metagenomic sequencing enables accurate pathogen and antimicrobial susceptibility profiling in complicated UTIs in approximately four hours

Anurag Basavaraj Bellankimath[1,8], Sverre Branders [1,8], Isabell Kegel[2,3], Jawad Ali [1], Fatemeh Asadi[1], Truls E. Bjerklund Johansen [4,5], Can Imirzalioglu[2,3], Torsten Hain[2,3], Florian Wagenlehner [3,6] & Rafi Ahmad [1,7] ✉

Urinary tract infections (UTIs) affect 405 million people worldwide. Current diagnostics rely on cultures, which can take 2 to 4 days. This study evaluates eleven culture-independent methods for sample preparation from 78 complicated UTI patients, followed by real-time nanopore sequencing and data analysis. The metagenomic results are highly consistent with culture-based clinical routines (MALDI-TOF/VITEK-2). The optimized method demonstrated an accuracy score of 99% (100/101) for pathogen identification and 90% (589/653) for antimicrobial susceptibility profiling with 95% specificity. The method's robustness is highlighted by its ability to accurately identify pathogens with as few as 32 bacterial cells/μL and a low bacterial-to-host cell ratio limit of 0.5. Additionally, mNGS identified 13 pathogens that routine diagnostics missed, which were subsequently confirmed by Vivalytic or PCR. This method is up to 30% more economical than published studies and commercial kits. DNA yield and flow cytometry can be used for pre-screening to reduce costs, which is crucial for clinical adoption. This research highlights the rapid diagnosis of clinical UTIs using a cost-effective and scalable method that requires around four hours from sample collection to informed decision-making. Furthermore, it aims to improve antimicrobial and diagnostic stewardship by reducing empirical treatment and ensuring more judicious antibiotic use.

Urinary Tract Infections (UTIs) are the second most prevalent infections worldwide, affecting 405 million people annually and causing significant morbidity and economic costs. Over the past three decades, global deaths due to UTIs have increased dramatically by 140%[1].

Untreated UTIs can progress to pyelonephritis[2] and urosepsis, which account for approximately one-quarter of all sepsis cases[3,4]. Catheter-associated UTIs account for 75–95% of hospital-acquired UTIs[5]. If not promptly treated, a secondary blood infection, particularly among

[1]Department of Biotechnology, University of Inland Norway, Hamar, Norway. [2]Institute of Medical Microbiology, Medical Microbiome-Metagenome Unit (M3U), Justus Liebig University Giessen, Giessen, Germany. [3]German Center for Infection Research (DZIF), Partner Site Giessen-Marburg-Langen, Giessen, Germany. [4]Institute of Clinical Medicine, University of Oslo, Oslo, Norway. [5]Institute of Clinical Medicine, University of Aarhus, Aarhus, Denmark. [6]Clinic for Urology, Pediatric Urology and Andrology, Justus Liebig University Giessen, Giessen, Germany. [7]Institute of Clinical Medicine, Faculty of Health Sciences, UiT—The Arctic University of Norway, Tromsø, Norway. [8]These authors contributed equally: Anurag Basavaraj Bellankimath, Sverre Branders. ✉e-mail: rafi.ahmad@inn.no

catheter-associated UTIs, is known to increase mortality rates to 30% posing a significant risk[6]. In severe UTIs, administering effective antibiotics as soon as possible is paramount.

A diverse range of virulence factors and pathogenic strains resistant to multiple drugs contribute to the complexity of UTIs[7,8]. Rapid and accurate identification of pathogens and determination of antimicrobial susceptibility are central to managing UTIs and improving antimicrobial stewardship in an era of increasing antimicrobial resistance (AMR). Furthermore, AMR may be attributed to polygenic traits, which cannot be adequately identified through exclusive amplification of individual resistance genes[9]. Given the advancing complexity of syndromic infections, polymicrobial infections, and AMR, the demand for multiplexed diagnostics has become increasingly important[10,11]. For culture-based methods, the time interval from sample collection to antibiotic susceptibility results is typically 2–4 days in the clinical routine[12,13].

Culture-free metagenomics next-generation sequencing (mNGS) using nanopore sequencing can detect the complete metagenome (in both mono and polymicrobial UTIs)[14] and known resistance mechanisms (AMR genes, SNPs, indels) present in a clinical sample[15]. mNGS is also superior in identifying fastidious, slow-growing pathogens and directly detecting AMR genes without the need for a targeted assay[16]. A recent meta-analysis on mNGS for UTI highlighted the excellent diagnostic performance for UTI, although only four of the 17 studies analyzed used the real-time Oxford Nanopore sequencing platform[17]. While these studies[14,18–21] highlight the potential of the Nanopore-based mNGS method for UTI, the challenges remain. A majority of current mNGS studies focus on acute, uncomplicated UTI cases, with very few samples on complicated cases. Moreover, most of these studies also utilize standard kit-based protocols with minimal modifications, such as IndiSpin pathogen kit (formerly the QIAamp pathogen mini kit), for the extraction of DNA without any host depletion[22]. The only exceptions are the studies by Liu et al.[18], in which the authors developed a wet lab pipeline, and Zhao et al.[19], who used a benzonase-based depletion method on 24 urine samples. These protocols also result in a total turnaround time of approximately 6–10 h if at all[14,18,21]. Furthermore, recent enhancements to the Nanopore sequencing chemistry have reduced error rates and increased throughput[23]. However, to date, no studies have evaluated this improved R10.4.1 chemistry for clinical UTI metagenomics. Another key limitation of existing studies is the poor prediction of antibiotic susceptibility. The current published works typically evaluate AST concordance by antibiotic class (e.g., β-lactamases)[24], if at all[18,25], thereby restricting its clinical use.

Clinical samples present significant challenges for implementing metagenomic pipelines due to variable pathogen loads and the high ratio of host to bacterial cells. Studies have determined the efficiencies of different extraction kits on patient samples, such as stool and blood[14,26,27]. However, efficient culture-independent methods for processing patient urine samples are rarely explored due to the low bacteria-to-host cell ratios and the presence of crystal salts, urea, and β-hCG, which complicate DNA extraction[28]. Processing to enrich target bacterial cells from patient specimens is crucial for developing effective sequencing-based diagnostics. The human genome has approximately 3.2 billion bases, while the average bacterial genome has only 5 million bases[29,30]. Thus, a typical human cell contains 1000 times more DNA than an average bacterial cell (6.4 pg v/s 5 fg), making the extraction of bacterial DNA from patient urine samples more complex due to the varying levels of host background[31,32]. Hence, the key challenge here is to develop a method that is accurate, rapid, and cost-effective, employing host depletion strategies to enhance pathogen detection using metagenomics and reliably diagnose UTI among the clinical samples. Such a culture and amplification-independent workflow could transform the treatment of complicated UTIs.

Here, we aimed to develop a rapid and cost-effective method for accurately diagnosing complicated UTIs (cUTI) in clinical settings. Our objective was to evaluate eleven distinct sample preparation methods for culture-free DNA extraction directly from the urine samples of patients with complicated UTIs by selectively depleting host cells and enhancing pathogen detection. This was followed by real-time nanopore sequencing and data analysis to identify pathogens and assess their antibiotic susceptibility.

## Results

We developed eight in-house methods designed to selectively deplete abundant host cells and enrich bacterial populations (Fig. 1 and Supplementary Table 1). We then compared these methods with three commercial test kits, starting with spiked urine samples containing relevant uropathogens, followed by urine samples from patients with cUTIs. The best-performing, optimized method was further evaluated using additional patient samples. All the mNGS results for the clinical samples were benchmarked against those from conventional culture-based results obtained with MALDI-TOF and VITEK-2 systems.

### DNA yield−the in-house method showed similar performance to the commercial kits when tested on spiked urine samples

Eight in-house methods, with combinations of either heat-liable salt activated endonuclease (hereafter referred to as HL_SAN) and medium salt activated endonuclease (hereafter referred to as M_SAN), were initially tested using urine spiked with *Escherichia coli*, along with four commercial methods as references (Supplementary Fig. 1 and Supplementary Table 1). Among the in-house methods, DNA extracted with the Naxtra Blood (hereafter referred to as NB) in HL_SAN and M_SAN combinations, on average, yielded higher DNA (5.9 μg) than Qiagen Blood & Tissue (hereafter referred to as BT) (3.5 μg). For the reference commercial methods, the NB kit (which does not deplete host DNA) achieved the highest DNA yield at 10.7 μg, while the Host Zero (hereafter referred to as HZ) kit had the lowest at 1.0 μg. The results from nanopore sequencing of the extracted DNA showed that >96.5% (Supplementary Fig. 1 and Supplementary Data 1) of *E. coli* reads were obtained across all sequencing methods. These findings demonstrated that the in-house method performed comparably to the commercial kits in terms of DNA yield and their compatibility and accuracy with nanopore sequencing.

### Host depletion−M_SAN shows slightly better performance in host depletion than HL_SAN

For selecting among the salt-activated nuclease (SAN) enzymes, qPCR assays were performed on depleted and undepleted samples to assess the extent of host depletion under different pH conditions using healthy urine samples spiked with *Enterococcus faecalis*. The results showed comparable depletion performance between M_SAN and HL_SAN, with M_SAN demonstrating a slightly higher depletion. Moreover, a slight loss of the spiked *E. faecalis* DNA was observed in the HL_SAN samples tested at pH 8.5 (Supplementary Fig. 2).

### The optimized method demonstrates superior host depletion efficiencies compared to commercial methods

With M_SAN performing slightly better than HL_SAN, we evaluated the overall method performance of the optimized method in depleting the host using the two commercial kits, MC5 and HZ. NB was not tested as it lacks a host depletion step. qPCR assays were conducted to quantify depletion efficiencies, expressed as fold change. Using the optimized method for the *E. coli* samples, we observed approximately $10^3$-fold depletion of human DNA, greater than HZ (approximately $10^2$-fold) and MC5 methods (approximately 10-fold). No loss of bacterial DNA was observed in samples extracted using the optimized method and HZ. However, a 2.2-fold loss of bacterial DNA (Average $\Delta Ct = 1.53$) was found between the depleted and undepleted samples for MC5 (Supplementary Fig. 3a). Similarly,

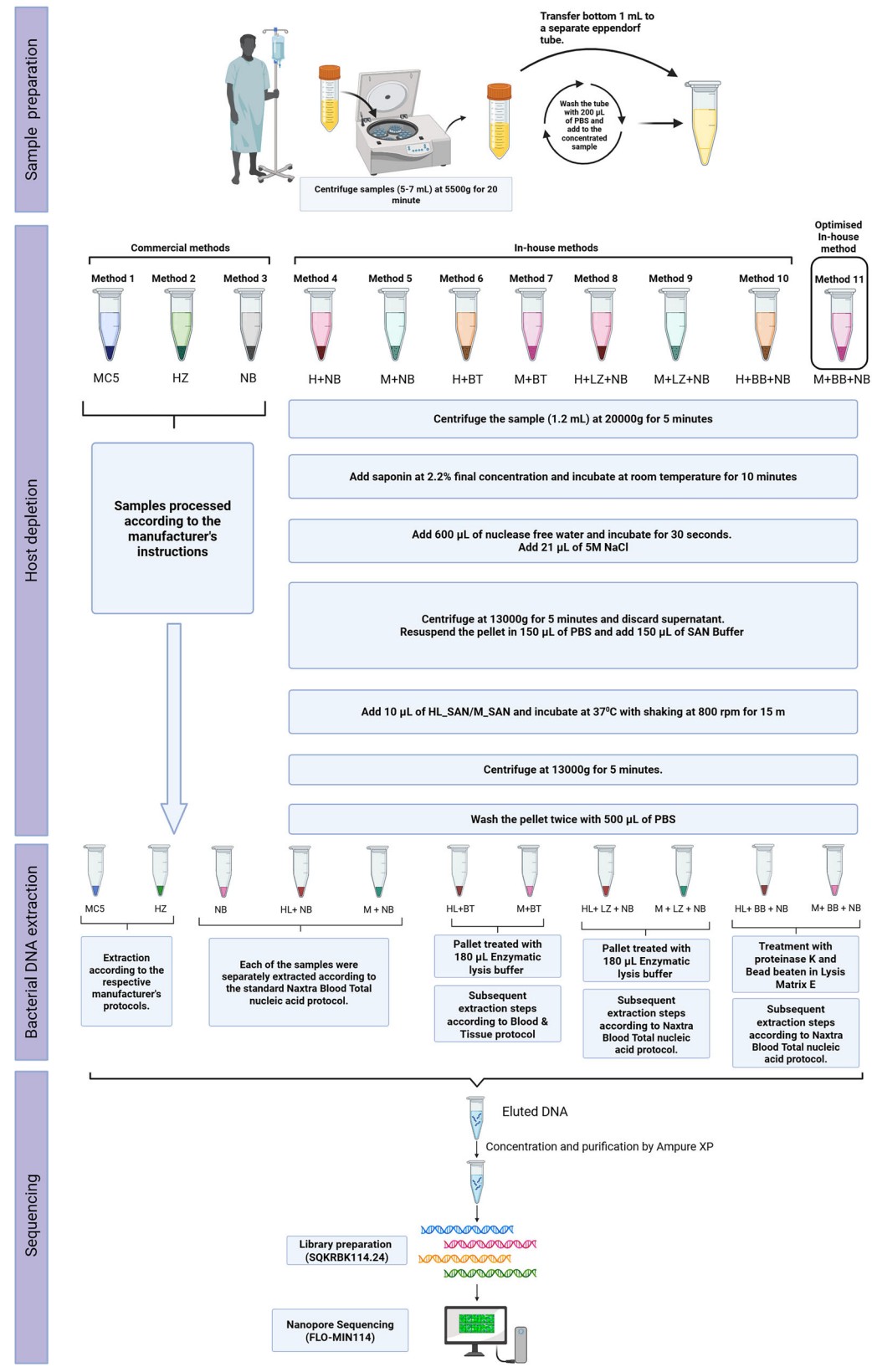

the optimized method outperformed others for samples spiked with *E. faecalis*, demonstrating a $10^2$-fold depletion of the host compared to HZ and MC5, which showed minimal to no significant depletion among those samples (Supplementary Fig. 3b). A slight loss of *E. faecalis* DNA was also noted in the samples extracted using HZ (Average ΔCt = 1.55) and MC5 (Average ΔCt = 2.93).

## Consistent host depletion performance of the optimized method across a variety of spiked clinical sequence types

As part of the final method validation, the optimized method was tested on a panel of common clinical uropathogenic sequence types to ensure its performance represents the species average. Comparing the depleted and undepleted samples (see Supplementary Fig. 4), the host

**Fig. 1 | Schematic overview of various sample processing methods for clinical samples.** Methods 1–3 are commercial, whereas methods 4–11 are SAN-based, in-house-developed techniques. Library preparation was performed using the rapid barcoding kit SQKRBK114.24, and the samples were sequenced on R 10.4.1 flow cells. The optimized method (M + BB + NB) performed best among the inhouse methods. For a detailed methodology breakdown, refer to the methods section of the manuscript. Abbreviations: MC5: Molysis Complete 5, HZ: Host Zero Microbial DNA kit, NB: Naxtra Blood Total Nucleic Acid kit, HL + NB: HL_SAN + Naxtra Blood Total Nucleic Acid kit, M + NB: M_SAN + Naxtra Blood Total Nucleic Acid kit, HL + BT: HL_SAN + Blood and Tissue kit, M + BT: M_SAN + Blood and Tissue kit, HL + LZ + NB: HL_SAN + Enzymatic Lysis + Naxtra Blood Total Nucleic Acid kit, M + LZ + NB: M_SAN + Enzymatic Lysis + Naxtra Blood Total Nucleic Acid kit, HL + BB + NB: HL_SAN + Bead Beating + Naxtra Blood Total Nucleic Acid kit, M + BB + NB: M_SAN + Bead Beating + Naxtra Blood Total Nucleic Acid kit. Created in BioRender. Branders, S. (2025) https://BioRender.com/l90b103.

Ct values increased by an average of 9.74 cycles, corresponding to approximately a $10^3$-fold depletion of the host. Bacterial Ct values showed slight enrichment in the depleted samples, with an average decrease of 1.6 cycles. The only exception was for samples spiked with *E. coli* ST1193, where both bacterial and host DNA were depleted in the depleted sample. These results demonstrate consistent performance of the optimized method in depleting the host across different species and sequence types, while maintaining bacterial DNA.

## Routine microbiological findings underscore the complexity of UTI samples

All 78 samples collected for method evaluation were leukocyte-positive, as determined by the routine dipstick test. The sample cohort consisted of approximately 38% female and 62% male patients. The cUTI sample cohort consisted of 65 (83%) samples, all of which were culture positive, and a control cohort of 13 (17%) samples, all of which were found to be culture negative. The positive samples were microbiologically complex, with 41% (32/78) classified as polymicrobial and 42% (33/78) as monomicrobial (Supplementary Data 2). Approximately 44% (14/32) of the polymicrobial samples contained three or more pathogens (Supplementary Fig. 5). Additionally, the samples exhibited complex resistance patterns.

Routine AST was performed on 55 individual samples, 89 bacterial pathogens, with 84% (75/89) of the tested pathogens being resistant to at least one antibiotic. Of those resistant pathogens, 51% (38/75) were multidrug-resistant, defined as resistant to three or more antibiotic classes. Therefore, a combination of both polymicrobial and multidrug-resistant UTI samples was evaluated across the methods.

## Benchmarking mNGS pathogen identification vs. standard urine culture (MALDI-TOF): optimized method is more effective than other in-house methods and comparable to commercial methods

The metagenomic results exhibited a strong correlation with routine clinical microbiology results in both pathogen identification and AMR detection (Table 1). The optimized method (M + BB + NB, method 11) demonstrated an overall accuracy score of 99% (100/101) across 53 samples (Table 1, Fig. 2a, c), constituting 88 pathogens correctly identified and 13 negative samples correctly identified. The optimized method showed a single false negative in sample 75, where *Proteus mirabilis* was not detected in a mixed infection with *Klebsiella pneumoniae*. In contrast, the non-optimized in-house methods demonstrated a mean accuracy score of 77% (49/64) for pathogen identification across 53 tested samples (Table 1, Fig. 2c and Supplementary Fig. 6), constituting 43 correct positive and 6 correct negative identifications, but missing (false negative) 10 bacteria and identifying 5 false positives. Methods 4 and 5 (see Supplementary Table 1) were unable to detect specific Gram-positive pathogens due to the inadequate performance of the NB method against these pathogens. The three commercial methods (MC5, HZ, and NB) had 81% (26/32) pathogen identification accuracy across 15 culture-positive and one culture-negative sample. 25 bacteria and one negative sample were correctly identified, but 6 false positives were found in addition to the routine results (Table 1, Fig. 2c, and Supplementary Fig. 6).

*E. coli* and *E. faecalis* were the most common pathogens identified in 33% (29/88) and 19% (17/88) of the samples. Three of the 26 (12%) polymicrobial samples tested with the optimized method contained both *E. coli* and *E. faecalis*. Two samples were found with *E. faecalis*−*K. pneumoniae* co-occurrence, and two samples had *E. faecalis*−*Aerococcus urinae* co-occurrence (Fig. 2d). Moreover, the optimized method identified 13 additional pathogens in 9 tested samples that routine identification had initially failed to identify (Fig. 2b). Routine testing identified three polymicrobial samples containing *A. urinae*, while mNGS detected *A. urinae* in four additional polymicrobial samples (41, 53, 54, and 62). mNGS identified *Actinotignum schaalii* in three polymicrobial samples (56, 57, and 62) where routine culturing did not. mNGS identified three additional pathogens that the traditional culture-based method missed: *Morganella morganii* in sample 41, *Providencia stuartii* in sample 59, and *Pseudomonas aeruginosa* in sample 70. A re-analysis of these samples by Vivalytic later confirmed the mNGS findings. This highlights the higher accuracy and in-depth pathogen identification potential of the mNGS approach.

## Benchmarking susceptibility testing against VITEK-2: optimized method outperforms other methods in predicting antibiotic susceptibility

For predicting antibiotic susceptibility, the optimized method reached an accuracy of 90% (589/653) (Table 1, Fig. 3a, c). In contrast, the non-optimized in-house methods and commercial methods achieved an accuracy of 88% (200/226) and 76% accuracy rate (170/225), respectively (Table 1, Fig. 3c and Supplementary Fig. 6).

For the optimized method, of the 88 bacterial pathogens, 51 had available routine AST data. 42 pathogens exhibited resistance to at least one antibiotic, accounting for 27 patient samples. Most false-negative/positive predictions were from the diaminopyrimidine, fluoroquinolones, and cephalosporins classes of antibiotics. Of these resistant pathogens, 14 were resistant to multiple (>2) antibiotics. The most commonly identified antibiotic resistance was against ampicillin (22 pathogens) (Supplementary Fig. 8). The most prevalent co-occurring resistance was ampicillin and amoxicillin/clavulanic acid (14 pathogens) (Fig. 3d). Ampicillin resistance often co-occurred with trimethoprim resistance (nine pathogens) in *E. coli*, *M. morganii*, and *P. aeruginosa*.

AcrAB-TolC was the most frequently observed resistance mechanism, identified in seven isolates (Supplementary Fig. 7). The most common co-occurrence of resistance mechanisms was EmrAB-TolC with MdtEF-TolC (five pathogens), all of which were found in *E. coli*. In addition to rapidly detecting pathogens and ARGs, mNGS detected relevant markers like virulence genes (e.g., *kdpDE* in *E. coli*, which was detected in samples 7, 9, 17, 23, 27, 37, 62, 67, and 68) and identified mutations associated with any known resistance mechanisms (e.g., *vanR-P* in *E. faecalis* was detected in sample 6), even for antibiotics not routinely tested. While the correlation of ARG data to phenotypic resistance remains challenging, these results highlight the ability to identify valuable information about resistance mechanisms and bacterial virulence, even in mixed samples.

## Comparison of method performance on the same sample

Methods 8–11 (see Supplementary Table 1), which included the lysozyme (H + LZ + NB and M + LZ + NB) and bead beating (H + BB + NB and M + BB + NB) modifications to address the shortcomings of NB. These methods, along with MC5 and HZ, were directly compared on a cohort

**Table 1 | An overview of the overall scoring metrics obtained by comparing mNGS testing to routine (MALDI-TOF/VITEK-2) testing on sample, pathogen, and antibiotic level for three categories of methods**

| Method | Total Samples | Per-sample accuracy | Total pathogens | Per-pathogen accuracy | Per-pathogen recall | Total antibiotics | Per-antibiotic accuracy | Per-antibiotic specificity |
|---|---|---|---|---|---|---|---|---|
| Optimized method | 53 | 100% (53/53) | 88 | 99% (100/101) | 99% (87/88) | 653 | 90% (589/653) | 95% (507/532) |
| In-house (non-optimized) | 36 | 100% (36/36) | 53 | 77% (49/64) | 81% (43/53) | 226 | 88% (200/226) | 100% (165/165) |
| Commercial | 16 | 100% (16/16) | 25 | 81% (26/32) | 100% (25/25) | 225 | 76% (170/225) | 96% (140/146) |

The optimized method (M + BB + NB), in-house (non-optimized) methods (H + NB, M + NB, H + BT, M + BT, H + LZ + NB, M + LZ + NB, H + BB + NB), and commercial methods (MC5, NB, HZ). The total number of samples tested, the total number of pathogens confirmed by routine testing, and the total number of available pathogen-antibiotic combinations tested in each category are given. Accuracy and recall scores are given as percentages along with the ratios of correct/total are given in brackets. Accuracy and recall calculations are described in detail in the Methods section for clarity.

of nine samples as shown in Supplementary Fig. 9. The optimized method was directly compared against non-optimized in-house methods (methods 8–10) and MC5 using six of these nine samples. Supplementary Fig. 10 shows accuracy metrics per method stratified by individual samples.

The optimized method achieved 100% accuracy (16/16), including 14 correct positive identifications and two correctly identified negative samples across 9 samples. The M + LZ + NB method had an overall accuracy of 91%, positively identifying eight bacteria and two negative samples correctly, but having missed one bacterium (false negative). The H + BB + NB method correctly identified six bacteria and two negative samples, but had three false negatives and three false positives, resulting in an overall accuracy of 57%. The H + LZ + NB method correctly identified four bacteria and two negative samples, but had one false negative and five false positives, resulting in an overall accuracy of 50%. The MC5 method was directly compared to the optimized method on six samples (see Supplementary Fig. 10) and correctly identified nine bacterial species and one negative sample, but had three false positives, resulting in an overall accuracy of 77%. The HZ method was compared to the optimized method on three samples and identified five bacterial species correctly, but also identified three false positives, resulting in an overall accuracy of 63%.

When comparing susceptibility testing, similar trends were observed (Fig. 3b). The optimized method achieved the highest accuracy score at 89% (79/89), followed by the MC5 method, which achieved 85% (62/73) accuracy. The optimized method has a specificity of 95.3% and did not overpredict resistance, while false negatives may be attributed to a lack of sequencing depth. The HZ method had an accuracy of 75% (42/56) and overpredicted resistance for six pathogen-antibiotic combinations in sample 81, where the optimized method and the MC5 method did not overpredict resistance.

### Approximately 19% of unclassified reads were recovered using MysteryMaster

After sequencing, as many as 19% (6220/32876) of sequencing reads remained unclassified after demultiplexing with Dorado. After processing the unclassified reads with the MysteryMaster demultiplex tool, it recovered, on average, 22% (89497/114749) of those unclassified reads (Supplementary Fig. 11).

Bacterial identification also revealed that switching the base-calling mode from FAST to HAC or SUP resulted in an average increase of about 20% (5085/4045) in identified bacterial reads. However, this increase did not influence the pathogen identification outcome. Since no meaningful difference was observed in the number of bacterial reads between SUP and HAC modes, the SUP accuracy mode may not be warranted for bacterial identification. Therefore, using higher accuracy base-calling modes can only be recommended if abundant computational resources are available and time is not a limiting factor.

### Strong relationship observed among flow cytometry data, DNA yield, and mNGS reads

Flow cytometry data (Supplementary Data 3) about the number of bacterial and human cells (erythrocytes, leukocytes, round, and squamous epithelial cells) from clinical samples were used to estimate the host depletion by correlating cell numbers with the number of bacterial and human reads (Supplementary Fig. 12). The method's robustness is demonstrated by its ability to accurately identify pathogens and predict AST in samples with a minimum bacterial-to-host cell ratio of 0.5 (Supplementary Data 1). Furthermore, sequencing data indicate that, on average, less than 1% of host reads were mitochondrial DNA, indicating depletion of both mitochondrial and host genomic DNA.

An AUROC analysis was conducted to evaluate whether total DNA yield and/or the number of bacterial cells quantified through

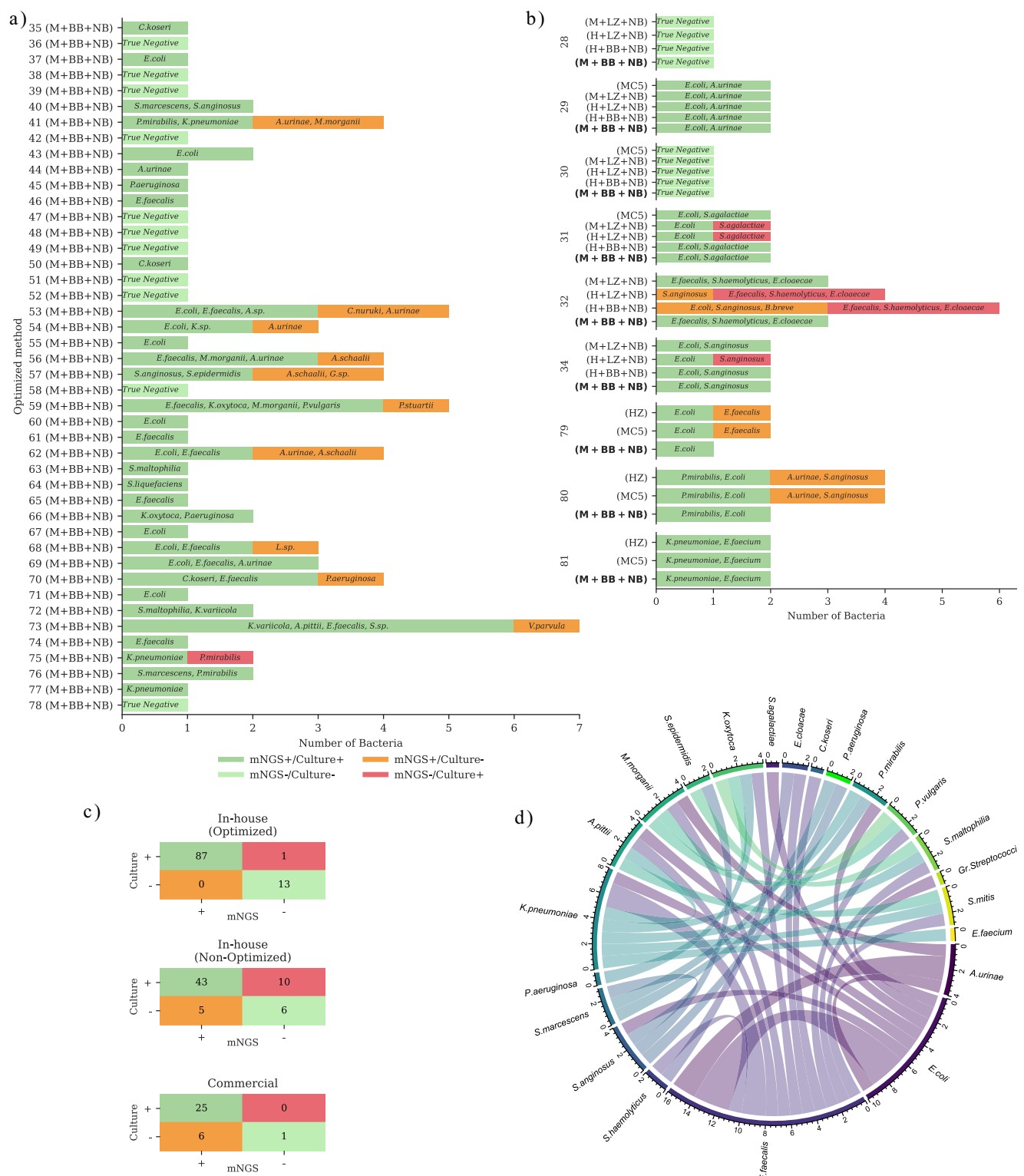

**Fig. 2 | Overview of pathogen identification benchmarking.** Subfigures (**a**) and (**b**) present benchmarking of species identification by mNGS against routine testing (MALDI-TOF). Dark green bars represent species identified through both routine testing and mNGS (concordant). Light Green bars indicate the samples found to be negative through both routine testing and mNGS (concordant). Orange bars highlight those species uniquely identified by mNGS, and red bars denote species uniquely identified by routine testing. Species names were annotated in the bars, and the x-axis indicates the number. Subfigure (**a**) shows samples analyzed using the optimized method (M + BB + NB). Subfigure (**b**) displays samples for which concordance with the routine of multiple methods were directly compared on the same samples. Subfigure (**c**) features confusion matrices comparing mNGS to the routine results stratified by the three different categories; The in-house optimized method (M + BB + NB), the in-house non-optimized methods (M + NB, H + NB, H + BT, M + BT, M + LZ + NB, H + LZ + NB, H + BB + NB), and commercial methods (NB, MC5, HZ). Subfigure (**d**) shows a chord diagram depicting the co-occurrence of pathogens in polymicrobial samples processed via the optimized method. Each species sector is annotated with absolute numbers to clarify the number of samples in which each species occurs. The size of the connections between sectors (the ribbon's thickness) reflects how frequently the connected species appear in a common sample. Colors are used to visually distinguish these connections. Source data are provided as a Source Data file.

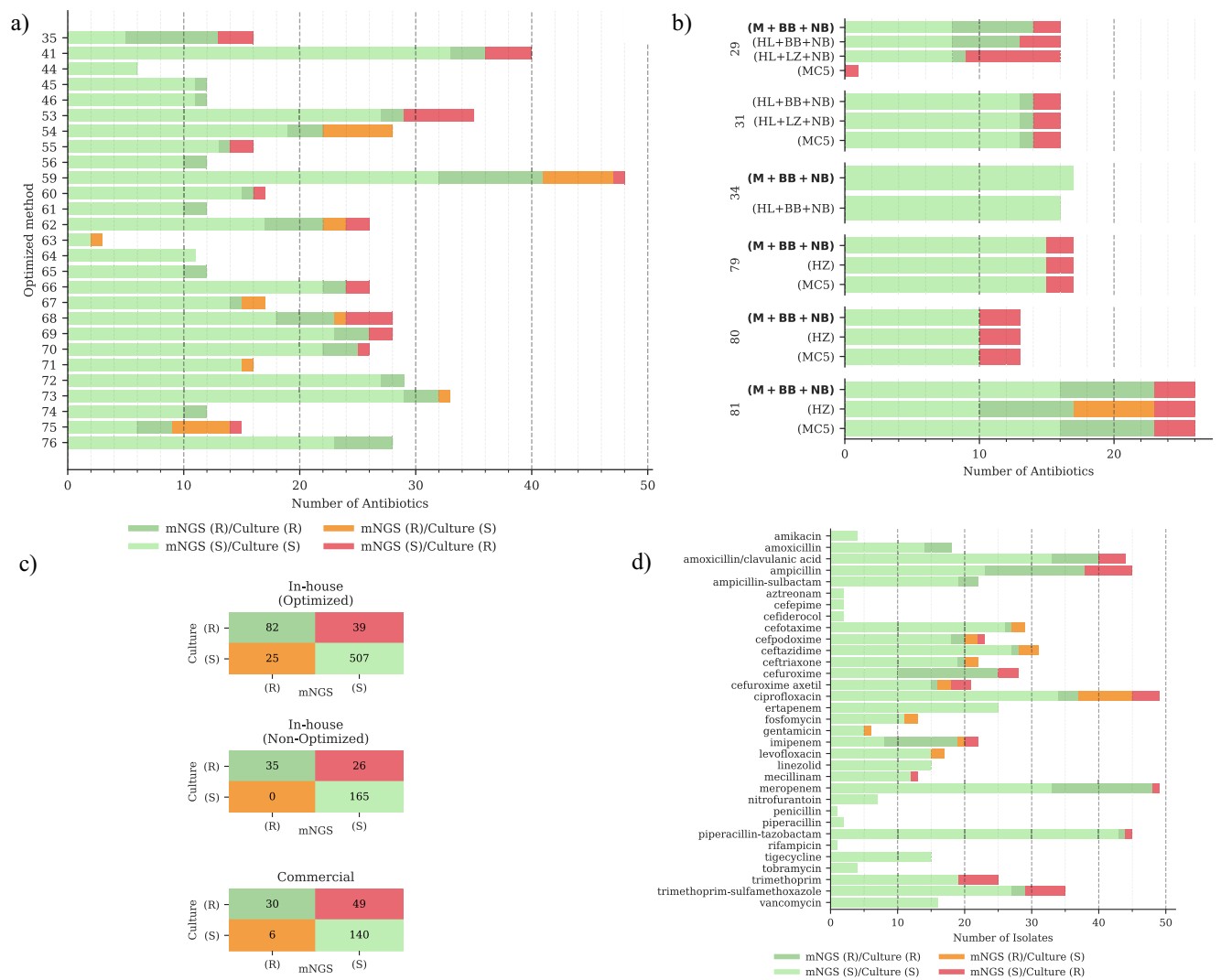

**Fig. 3 | Overview of antimicrobial susceptibility benchmarking.** Subfigures (**a**), (**b**), and (**d**) present concordance between mNGS results and routine AST stratified by samples, methods, and antibiotics categories. Dark green bars indicate that both routine AST and mNGS results suggest resistance, whereas light green bars indicate that both routine AST and mNGS indicate susceptibility. Red indicates cases where resistance was identified in routine AST, but no corresponding resistance mechanism was detected through mNGS (false negative). Orange bars denote detected ARGs without phenotypic resistance to the corresponding antibiotic. Subfigure (**a**) illustrates the results for samples extracted using the optimized method (M + BB + NB). Subfigure (**b**) displays samples for which concordance with routine AST for multiple methods was directly compared on the same samples. Subfigure (**c**) features confusion matrices comparing resistant/susceptible identification through mNGS (predicted) to the routine AST results (VITEK-2) stratified by the three different categories: The in-house optimized method (M + BB + NB), the in-house non-optimized methods (M + NB, H + NB, H + BT, M + BT, M + LZ + NB, H + LZ, H + BB + NB), and commercial methods (NB, MC5, HZ). Source data are provided as a Source Data file.

flow cytometry could serve as pre-screening indicators to differentiate between culture-positive and culture-negative samples (Fig. 4). The analysis of DNA yield (Fig. 4a) resulted in an AUROC of 0.87 (95% CI: 0.57–0.98). A cutoff value of 273 ng provided a positive predictive value (PPV) of 0.97 (95% CI: 0.88–1.00) and a negative predictive value (NPV) of 0.53 (95% CI: 0.31–0.90). In comparison, the assessment of bacterial cell counts (Fig. 4b) produced an AUROC of 0.90 (95% CI: 0.78–0.98), leading to a PPV of 0.97 (95% CI: 0.92–1.00) and an NPV of 0.67 (95% CI: 0.32–1.00) at a cutoff of 451 cells. The AUROC values indicate that both metrics are reasonable predictors of culture positivity.

Moreover, the results imply a detection threshold of approximately 500 cells/µL, correlating with a positive predictive rate exceeding 90%. Further analysis indicated these cutoffs could be improved when stratified by sample origin (Supplementary Fig. 13), i.e., whether the sample was a midstream urine sample or other (catheter, exprimate urine, first stream urine,

etc). DNA yield produced an AUROC of 0.83 and 0.88 for midstream and other urine samples, respectively (Supplementary Fig. 13a). The optimal cutoff values were 241 ng (PPV 0.96, NPV 0.71) and 18 ng (PPV 1.0, NPV 0.64). Bacterial cell counts (see Supplementary Fig. 13b) produced an AUROC of 0.93 and 0.83 for midstream and other urine samples, respectively. The optimal cutoff values were 254 cells (PPV 0.96, NPV 1.0) and 39 cells (PPV 1.0, NPV 0.5). CFU/mL is used for the determination of culture positivity in the clinical routine. A figure of CFU/mL AUROC was included in Supplementary Fig. 14.

**Around four hours of turnaround time (TAT) from initiating sample analysis to pathogen identification and ARG detection**
Testing the optimized method (M + BB + NB) on 53 clinical samples demonstrated that this method achieved the highest accuracy scores among all in-house developed methods, with 99% (100/101) for pathogen identification and 90% (589/653) for

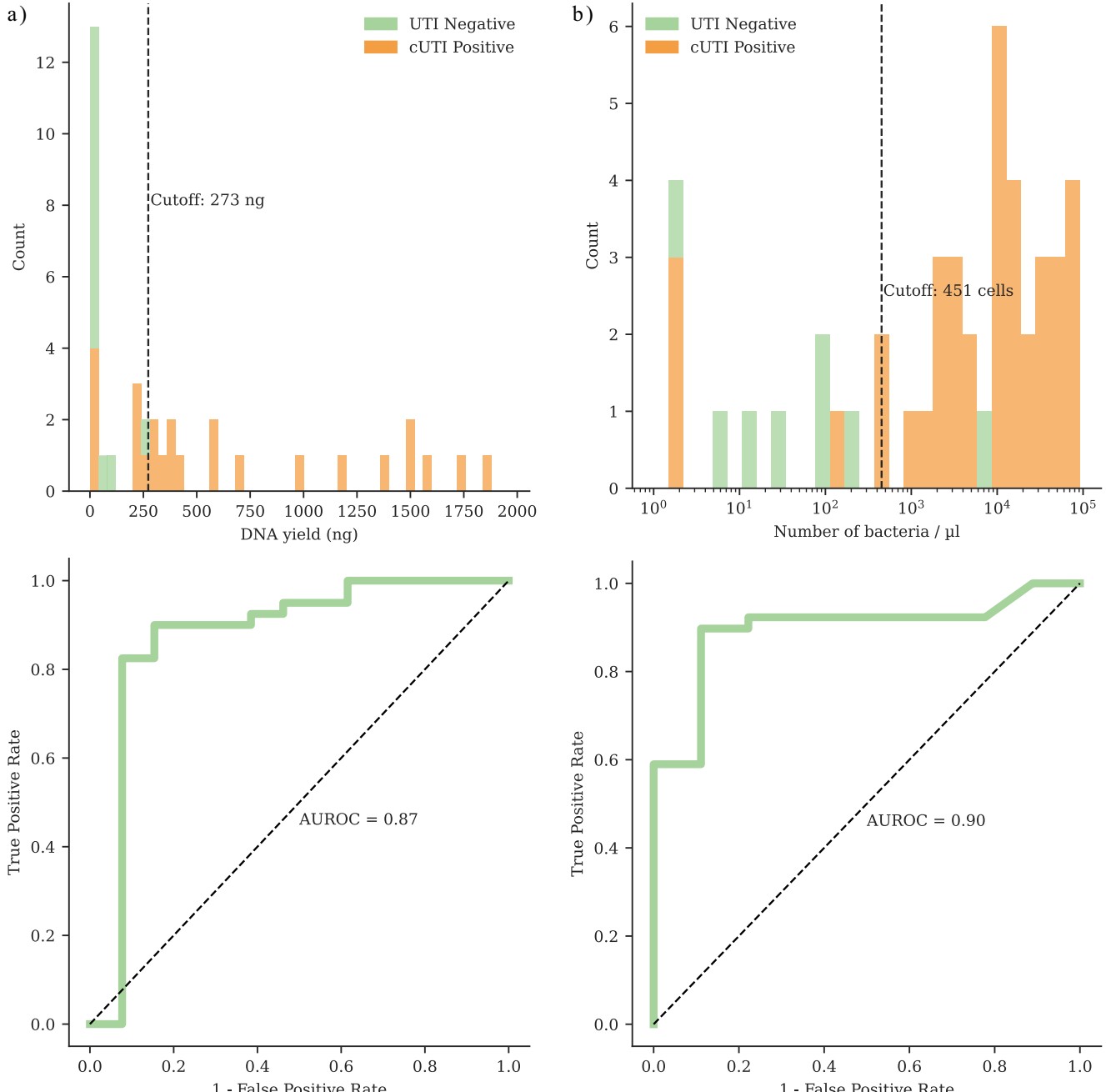

**Fig. 4 | DNA yield and bacterial cell receiver operating characteristic (ROC) curve analysis.** The analysis was conducted on samples extracted using the in-house optimized method ($n = 53$). Subfigure (**a**) presents the results for DNA yield, while (**b**) displays the results for the number of bacterial cells/µL as identified through flow cytometry. The histogram (top) illustrates the distribution of each variable for culture-positive and negative samples, highlighting the optimal cutoff to distinguish between culture-positive and negative samples. The ROC curve (bottom) is annotated with the area under the ROC curve (AUROC) for each variable. The AUROC values of 0.87 (95% CI: 0.57–0.98) and 0.90 (95% CI: 0.78–0.98) demonstrate that both DNA yield and bacterial cell counts are reliable predictors of culture positivity. Source data are provided as a Source Data file.

antimicrobial susceptibility predictions, making it the most effective extraction method. This method identified pathogens within 2–25 min of sequencing, while resistance genes were detected in as little as 3 min to a few hours. The turnaround time for the optimized method, which includes host depletion, DNA extraction, library preparation, nanopore sequencing, and real-time analysis, was approximately 4 h (Fig. 5). This TAT was determined from the time the sample was received at the lab for analysis. In comparison, the commercial methods had turnaround times of approximately 4.5 h for MC5 and approximately 6 h for HZ (Supplementary Fig. 15).

**The optimized method is also the most cost-effective**

The method costs approximately $36 per sample, which includes host depletion & DNA extraction ($5.80), library preparation ($5.60), and sequencing with a barcoding kit for 24 samples ($25 - when purchasing a 24-flow cell pack, FLO-MIN114). These costs are lower than those of the commercial kits MC5 ($51) and HZ ($48) (Supplementary Data 4). The sequencing cost was $30.60, covering the MinION flow cell and the rapid barcoding kit for 24 samples[33]. This cost remained consistent across all methods, while the DNA extraction costs varied. The costs can be significantly reduced by pre-screening the samples based on the DNA yield and bacterial cell count

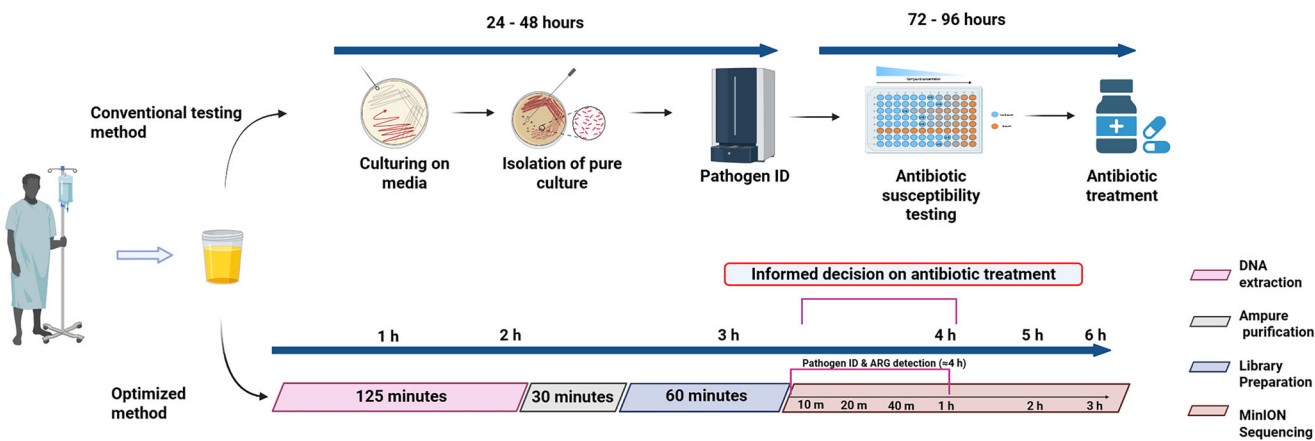

**Fig. 5 | An overview of the timeline for pathogen and ARG detection using the in-house optimized method (M + BB + NB) compared to conventional routine diagnosis.** The timeline for the method is organized and color-coded according to the specific steps involved: DNA extraction, purification, library preparation, and sequencing. This method has turnaround time (TAT) of approximately four hours from the sample receipt for detecting the pathogens and making informed decisions about antibiotic treatment. Created in BioRender. Branders, S. (2025) https://BioRender.com/qsw6kpk.

cutoffs obtained from the AUROC analysis (Fig. 4) since DNA extraction costs only $5.80.

## Discussion

UTIs are among the most common bacterial infections, but diagnosing them can be challenging due to their complex etiology and clinical presentations. Standard urine culture (SUC) has been the routine method for diagnosing UTIs for decades; its lengthy turnaround time and overall sensitivity are significant limitations[34,35]. Unlike the SUC, metagenomics is a target-agnostic approach that facilitates rapid and targeted antibiotic treatment of UTIs[36]. In this study, we aimed to develop a rapid, selective, and cost-effective host depletion-based metagenomic sample preparation, nanopore sequencing, and data analysis workflow for analyzing UTI samples. The development involved comparing the performance of these methods with commercial alternatives. Eight in-house and three commercial methods were tested on complex UTI samples, after which the top-performing method was further evaluated using additional samples. The mNGS results were benchmarked against routine clinical results obtained from the MALDI-TOF and VITEK-2 systems for pathogen identification and antimicrobial susceptibility, respectively.

In-house methodologies were developed that combine saponin and SAN endonucleases for host depletion, along with magnetic bead-based (NB) or column-based DNA extraction (BT) methods. In the initial qPCR-based studies on spiked urine samples, M_SAN showed a slight advantage over HL_SAN in host depletion, likely because M_SAN functions best in a neutral to slightly basic pH environment, whereas HL_SAN requires a fully basic pH for optimal performance[37,38]. The magnetic bead-based NB method was preferred for DNA extraction due to several advantages over the BT method. Advantages include longer average read lengths (1957 bp compared to 1293 bp), shorter assay times (45 min versus 78 min), lower per-sample extraction costs ($3.30 versus $4.10), and the potential for more straightforward automation using systems such as the KingFisher Flex. However, the NB method faced challenges with Gram-positive bacteria, prompting improvements in the lysis steps by incorporating enzymatic or bead-beating procedures in each SAN + NB variant. Furthermore, this optimized method outperformed the commercial kits MC5 and HZ in depleting host DNA and enriching *E. coli* DNA in the spiked urine samples. The optimized method achieved approximately a $10^3$-fold host depletion, compared to a $10^2$-fold depletion with HZ and a 10-fold depletion with MC5. A comparable outcome was observed with samples spiked with *E. faecalis*. Bacterial morphotypes significantly influence DNA extraction, with bacterial morphology and cell wall

composition having a substantial impact on lysis and bacterial DNA recovery[39]. Uropathogens such as *E. coli*, *E. faecalis*, *P. aeruginosa*, *P. mirabilis*, etc. are known to exhibit morphological plasticity, particularly under stress[40]. Therefore, the optimized method was subsequently tested on a panel of six prevalent clinical uropathogens with different sequence types to assess its performance across various morphotypes. The results show an average $10^3$-fold depletion across the pathogenic sequence types and species without the loss of spiked pathogenic DNA. The only deviation was observed in *E. coli* ST1193, where both the *E. coli* DNA and that of the host were depleted. Hence, the optimized method consistently performed across all spike-in experiments involving different species of clinical uropathogens and sequence types, thereby confirming its suitability across divergent species that could potentially exhibit different morphotypes.

Eleven methods, including eight in-house methods and three commercial kits (see Fig. 1), were evaluated using 78 complex UTI patient samples. The urine samples were everyday clinical specimens collected as midstream and catheter urines from patients at a reference hospital for complicated, severe UTIs. The samples were microbiologically complex, with 41% (32/78) being polymicrobial and 42% (33/78) monomicrobial, and 44% (14/32) of the polymicrobial samples contained three or more pathogens. Additionally, 84% (75/89) of the tested pathogens were resistant to at least one antibiotic, and 51% (38/75) were multidrug-resistant. AST prediction was performed against a panel of 39 UTI antibiotics. To our knowledge, no other study has evaluated such a wide variety of complex UTI samples. In the study by Liu et al., only 4.4% of the samples were polymicrobial, containing just two pathogens. Furthermore, published mNGS studies have focused on AST prediction or are limited to specific antibiotic classes and pathogens[18,21,41].

The pathogen identification accuracy for the optimized method was 99%, the in-house non-optimized method was 77%, and the commercial methods achieved 81%. The optimized method (M + BB + NB) detected all but one pathogen among the 53 tested clinical samples, correctly identifying a total of 87 pathogens. This method accurately identified pathogens in highly complex samples containing up to five unique pathogen species and a significant number of host immune cells. The method's accuracy also compares well with existing studies on clinical samples. Pathogen identification concordance between the optimized method and clinical routine results ($n = 78$, accuracy 99%, precision 100%, recall 99%) was higher than results reported in comparable studies, while turnaround time was the shortest reported at only 4 h approximately: Liu et al. $n = 1327$ accuracy 95% (precision 64%, recall 88%)[18], Zhang et al., $n = 75$ accuracy 92% (precision 98%, recall

87%)[21], Janes et al., n = 81 accuracy 89% (precision 87%, recall 79%)[24], Jia et al., *n* = 43 accuracy 77% (precision 34%, recall 50%)[41].

Furthermore, it identified additional pathogens in polymicrobial samples. Interestingly, *A. urinae* and *A. schaalii* were identified in polymicrobial samples via mNGS five and three times, respectively, whereas routine cultures failed to detect them. These pathogens are often underreported due to their fastidious nature, obscure role in infections, and the challenges they present in identification, particularly within polymicrobial samples, leading to their frequent dismissal as irrelevant contaminants[42–46]. As diagnostic tools advance, the significance of pathogens such as *A. urinae* and *A. schaalii* may change over time. Consequently, the method identifies this species; however, the physician will determine the best treatment approach in each case. Therefore, these samples were re-evaluated using the Vivalytic analyzer, confirming their presence. This highlights the power of the optimized method in detecting difficult-to-culture pathogens.

Forty-one percent of the total samples included in the study were known to be polymicrobial infections. The high number of polymicrobial samples can be attributed to the sample types, as cUTIs are more likely to be polymicrobial compared to uncomplicated UTIs[47]. Further analysis of the co-occurrence of resistance and pathogens in these samples revealed a significant co-occurrence of *E. coli and E. faecalis*. This observation aligns with prior research, demonstrating that these two species frequently co-exist in polymicrobial UTI cases, especially in catheterized patients, where they co-localize within a biofilm on catheters and employ dynamic adaptive strategies that promote their coexistence[48,49]. Furthermore, the sequencing results indicate low levels (<1% of host reads) of mitochondrial DNA, suggesting that the mitochondrial membrane is also lysed, resulting in DNA depletion. Previously published methods[35] have struggled with selectively depleting mitochondrial membranes, leading to a significant amount (88.8%) of sequencing data of mitochondrial origin. Our optimized strategy has also successfully addressed this limitation.

The overall accuracy of antibiotic susceptibility predictions varied among methods, with the optimized method achieving an accuracy of 90% compared to 76% for the commercial methods and 88% for in-house methods. The optimized method achieved AST concordance for 653 antibiotic pathogen combinations, including highly complex polymicrobial samples. Among the samples with available routine AST data, 82% (42/51) were resistant to at least one antibiotic, and 27% (14/51) exhibited resistance to multiple antibiotics. Ampicillin resistance was found to frequently co-occur with resistance to amoxicillin/clavulanic acid or trimethoprim. It has been suggested that co-selection via ampicillin and amoxicillin is a key factor driving trimethoprim resistance[25]. The most commonly identified resistance mechanisms among resistant samples were AcrAB-TolC and EmrAB-TolC. AcrAB-TolC was frequently found to co-occur with MdtEF-TolC. AcrAB is constitutively expressed in *Enterobacteriaceae*. EmrAB and MdtEF are differentially expressed under stress conditions, such as acidic environments, which may exert selective pressure on all three proton-motive force-driven efflux pumps[50–52].

AST predictions related to cephalosporins, quinolones, and diaminopyrimidines were the primary contributors to false predictions (Supplementary Fig. 7). This was mainly linked to the detection of *ampC* variants and the efflux pumps AcrAB-TolC and MdtEF-TolC. Most of the detected *ampC* variants (*blaEC-5, blaEC-8, blaEC-15, blaEC-18, and blaEC-19*) were found in samples of *E. coli*. Although *E. coli* naturally possesses the chromosomal *ampC* gene, the absence of the *ampR* gene results in reduced expression[53]. The low levels of expression of the *ampC* gene in *E. coli* may result from mutations in the promoter and attenuator regions rather than induction due to exposure to antibiotics such as β-lactams[54]. Hence, their detection rarely leads to relevant detectable resistance. Similarly, detecting efflux pumps AcrAB-TolC and MdtEF-TolC alone rarely results in significant phenotypic resistance[55]. In addition to detecting ARGs, mNGS

identified relevant virulence genes such as *kdpDE* in *E. coli*. This is a two-component regulatory system that contributes to the activation of virulence factors and facilitates the intracellular survival of pathogenic bacteria, such as *Staphylococcus aureus* and *E. coli*[56,57]. The method identified mutations linked to known resistance mechanisms like *vanR-P*[68] in *E. faecalis*, even for antibiotics not routinely tested. Despite the challenges in correlating ARG data to phenotypic resistance, these findings highlight the potential to uncover important information regarding resistance mechanisms and bacterial virulence, even in polymicrobial samples.

The current clinical routine diagnostics of UTI takes 2–4 days, necessitating a lengthy regimen of empirical antibiotic treatment in severe cases. While a mNGS-based detection method with a TAT of as little as 5 hours has been proposed[28], it was tested on spiked urine samples rather than clinical ones. Previously, Illumina-based mNGS methods have been tested on clinical UTI samples, but the sequencing time of these methods takes at least one day[24,41,59]. Nanopore-based methods are more rapid, and such methods with TATs ranging between 6 and 10 h have been successfully tested on clinical UTI samples[18,21]. The current study further improves upon this, with the optimized method having a total TAT of only 4 h. Timely treatment with the appropriate antibiotic is essential for effectively managing severe infections, such as urosepsis and surgical cases, as well as promoting antibiotic stewardship.

Cost-effectiveness is of vast importance when adopting technology that improves the treatment of patients and antimicrobial stewardship. The optimized method is affordable, costing approximately $36 and being 30% more economical than commercial kits, making it ideal for low-resource settings. This cost includes host depletion, DNA extraction, library preparation, and sequencing expenses for 24 multiplexed samples. When handling larger sample sizes, sequencing costs can be further reduced by multiplexing up to 96 samples and sequencing on a PromethION flow cell, lowering the cost to $15[33,60]. To our knowledge, this is the only study on UTI mNGS that reports the cost aspects of its method or compares it to commercial methods, making it a novel aspect. The overall per-sample costs can be further reduced by utilizing flow cytometry-based cell counts and DNA yield cutoffs. Previous studies have demonstrated that flow cytometry can detect significant bacteriuria in patient urine samples, but lacks the capability to identify the pathogen[61,62]. In this study, we built upon this observation and assessed the feasibility of using bacterial cell counts from flow cytometry data and DNA yield as pre-screening criteria to differentiate between culture-positive and negative samples. The flow data were used for the AUROC calculations because it accurately reflects the sample state from which the DNA extraction was performed. The AUROC analysis showed 0.90 (95% CI: 0.78–0.98) for bacterial cell count and 0.87 (95% CI: 0.57–0.98) for DNA yield, indicating that both flow cytometry and total DNA yield can serve as indicators for distinguishing between positive and negative samples. Moreover, the flow cytometry was performed in an automated mode in parallel to the conventional and metagenomic analysis, and hence, does not add to the turnaround time. This method could lower costs to approximately $6 by removing the need for sequencing negative samples. This metric will be further validated in a follow-up study using over 200 clinical urine samples.

A limitation of the developed host depletion protocols is their inability to detect fungal pathogens. The lysis effect of saponin is mediated through interaction with sterol groups present in eukaryotic cell membranes but not in bacterial cell membranes[63,64]. Unlike bacteria, fungal cell membranes contain sterol ergosterol, which increases their susceptibility to saponin and subsequent osmotic stress[65]. Therefore, a revised protocol should facilitate the detection of fungal pathogens. The optimized method was tested on a limited number of samples in the present study. It is currently being evaluated with an

additional 200+ clinical urine samples to assess its robustness and improve the analytical metrics.

In summary, this study demonstrates the effectiveness of our optimized DNA extraction method (M + BB + NB), enhanced by nanopore sequencing and real-time data analysis, for the rapid diagnosis of UTIs in clinical settings. Overall, the optimized method had a pathogen identification accuracy of 99% and an antibiotic susceptibility prediction accuracy of 90% with a limit of detection of $10^3$ CFU/ml and a bacteria-host ratio of 0.5. Therefore, this approach achieves an overall accuracy rate exceeding 90%, which is typically essential for implementing new diagnostic techniques in clinical microbiology[66,67]. Furthermore, these mNGS techniques can provide deeper insights into the mechanisms of resistance. An accurate, scalable, easy-to-implement, and cost-effective rapid point-of-care method can improve clinical management of time-sensitive conditions, including urosepsis and kidney transplant infections. Additionally, it will strengthen antimicrobial and diagnostic stewardship by shortening the duration of empirical treatment and ensuring more judicious antibiotic use. By reducing the diagnostic timeframe from four days to four hours, this approach could prevent the unnecessary use of 1.62 billion daily doses (405 million cases × 4 days) of empirical broad-spectrum antibiotics each year before initiating personalized, targeted therapy against an identified pathogen. This will not only optimize UTI treatment but also enhance antimicrobial stewardship and mitigate further development of antimicrobial resistance.

## Methods

### Study design

We developed eight methodologies for complex UTI diagnostics using real-time nanopore sequencing and data analysis (Supplementary Table 1). Eleven methods, including eight combinations of SAN endonucleases (HL_SAN/M_SAN) and three commercial kits (Fig. 1), were evaluated, with a total of 78 patient samples collected from the urology department. The study was conducted in three phases: a method development phase involving spiked samples, followed by the clinical evaluation of the developed methods in two testing steps, and finally, the best-performing method was tested on additional samples. The best-performing method was then selected based on host depletion, cost, accuracy, and turnaround time. This method was further validated on an additional 53 samples. All clinical evaluations were performed by benchmarking against routine data obtained from MALDI-TOF and VITEK-2 systems.

### In-house method development with spiked urine

A saponin and SAN endonucleases (HL_SAN/M_SAN)-based assay was developed and tested on spiked urine samples. Saponin and SAN have been previously described for respiratory samples, including sputum, bronchoalveolar lavage, and endotracheal secretions[68]. The 3−8 mL urine sample was centrifuged at 5500×g for 20 min at room temperature, after which the supernatant was discarded, leaving behind 1 mL at the bottom. After resuspending the bottom 1 mL and any possible pellet in an Eppendorf tube, the tube was rinsed with 200 μL of PBS. The resulting 1.2 mL sample was centrifuged at 20,000 × g for 5 min, and the top 500 μL of clear supernatant was discarded. The remaining 700 μL was used for subsequent steps. Then, 700 μL of 4.4% saponin solution (Sigma Life Sciences) was added to the sample, and the mixture was incubated in an Eppendorf ThermoMixer block for 10 min at 800 rpm. Next, 600 μL of nuclease-free water was added to the mixture, vortexed, and incubated at room temperature for 30 s, followed by the addition of 21 μL of 5 M NaCl to induce osmotic shock and lyse host cells. The mixture was then centrifuged at 13,000 × g for 5 min, after which the supernatant was discarded, and the resultant pellet was resuspended in 150 μL of PBS. Subsequently, 150 μL of SAN buffer (5.5 M NaCl, 0.1 M MgCl₂) was added to the resuspended pellet with 10 μL of either HL_SAN

(25 U/μL, Arctic-Zymes Technologies, Tromsø, Norway) or M_SAN (29.2 U/μL, Arctic-Zymes Technologies, Tromsø, Norway). The mixture was incubated at 37 °C for 15 min at 800 rpm in an Eppendorf ThermoMixer. The samples were centrifuged at 13,000 × g for 5 min, after which the supernatant was discarded, and the pellet was washed twice with 500 μL of PBS. The obtained pellet was then used for extracting microbial DNA. Following host depletion, the microbial DNA was extracted using the Naxtra Blood Total nucleic acid kits (LSBL0048, Lybe Scientific AS, Norway, here referred to as NB) or the DNasey Blood and Tissue kit (cat: 69504, Qiagen, Germany, referred to as BT). The extractions were performed according to the respective manufacturer's instructions for both kits, and the Gram-positive pre-processing steps for BT were followed during the extractions. The subsequent extraction steps for BT, including the Gram-positive pre-processing steps, took ~75 min, while the NB extraction steps took ~40 min.

### Commercial kits used as reference

In addition to the developed SAN assays, four different commercial DNA extraction kits, Molysis Complete 5 (D-321-050, Molzym GmbH, Germany, here on referred to as MC5), Host Zero microbial enrichment kit (D4310, Zymo Research, USA, here on referred as HZ), and NB, were evaluated as standalone kits. The experiments were conducted according to the standard kit protocol without modifications.

### DNA yield and nanopore sequencing testing of the methods with spiked urine

For initial method development, healthy urine samples were spiked with *Escherichia coli* NCTC 13441 at $10^8$ and $10^5$ CFU/mL concentrations according to the previously established spiking protocols[28]. Dipsticks were used to confirm the healthy urine samples. Initially, the spiking was carried out in 5 mL of urine, but the volume was later increased to 10 mL. After spiking, the initial SAN assays included a 10% EDTA pretreatment step to prevent crystallization and aid in DNA extraction. The extractions were performed with all eight SAN combinations (four SAN combinations extracted with NB and four SAN combinations extracted with NB pre-treated with 10% EDTA) and four commercial kits. Spiked pathogenic *E. coli* DNA was verified using PCR, as given in. A subset of the extracted DNA samples (commercial kits and HL_SAN/ M_SAN methods extracted with NB) was sequenced on MinION R10.4.1 flow cells.

### Comparison of the depletion efficiencies between HL_SAN and M_SAN endonucleases

Urine obtained from healthy donors was confirmed by a negative dipstick test and then spiked with WBC from the buffy coat to mimic clinical samples. Prior to spiking, the WBCs were counted using a hemocytometer and adjusted to get a cell count of $2 \times 10^4$ cells/mL, and the samples were spread plated to determine background contaminant and microflora[28]. It was subsequently spiked with *E. faecalis* to achieve a final concentration of $10^5$ CFU/mL. The WBC and bacterial spiking were performed on a batch of healthy urine to establish a common baseline across all the aliquots. Following spiking, DNA was extracted using H + BT and M + BT methods for test samples, whereas the undepleted control samples were extracted using the standard Blood and Tissue protocol. All extractions were performed in triplicate (n = 3), and the depletion efficiencies were determined through quantitative PCR (qPCR) cycle thresholds (Ct) of species-specific primers for host/human and *E. faecalis* in both depleted and undepleted samples. The extent of host depletion and bacterial enrichment was calculated using ΔCt, normalized using the Ct of the undepleted biological control. The Normalized Ct values were represented as fold changes calculated using equation $2^{-\Delta Ct}$. The results were further analyzed to determine p-values among different replicate groups using a 2-tailed unpaired *t* test.

## Comparison of the depletion efficiencies of the optimized method to commercial kits

The performance of the optimized method was benchmarked against two commercial kits, MC5 and HZ. Similar to the endonuclease experiments, healthy urine samples were spiked with WBCs isolated from the blood and a uropathogens at $2 \times 10^4$ cells/mL and $10^5$ CFU/mL, respectively, to mimic clinical samples. The experiment was conducted in two batches: one spiked with *E. coli* NCTC 13441 (ST131), and the other spiked with *E. faecalis* CCUG 9997 (ST30). Spiking was carried out on the pooled batch in each instance to maintain a uniform baseline. The spiked samples were then tested in triplicate using the three methods to assess host depletion and any unintended loss of bacteria. The extent of host depletion and bacterial enrichment was calculated using ΔCt, normalized using the Ct of the undepleted biological control. The fold change and the statistical analysis were performed as mentioned in the previous section.

## Testing of the optimized method on multiple sequence types of clinically prevalent uropathogenic species

The optimized method was tested on a panel of prevalent clinical uropathogens, including sequence types of *E. coli* (ST38 and ST1193), *K. pneumoniae* (ST15 and ST147), *E. faecalis* (Two strains of ST6), *Proteus mirabilis* (ST1), and *P. aeruginosa* (ST111 and ST235) to assess its performance across different clinical species and morphotypes. These clinical isolates and the specific sequence types were selected based on their prevalence in UTIs[69–73]. The only exception is for *P. mirabilis*, where only one isolate was available in-house. A batch of healthy urine samples was spiked with WBCs isolated from whole blood at $5.84 \times 10^4$ cells/mL and with the uropathogens at $10^5$ CFU/mL. SYBR Green-based qPCR assays were used to compare the level of host and spiked pathogen in both depleted and undepleted samples, as described in the qPCR section.

## PCR and qPCR assays

SYBR Green-based qPCR assays were performed to estimate host and bacterial DNA by targeting the human β-actin gene and the respective bacterial genes. The following genes were used for each pathogen: *E. coli* (*UspA*), *E. faecalis* (*GroES*), *K. pneumoniae* (*khe*), *P. mirabilis* (*UrePa*), and *P. aeruginosa* (*phzA2*) (Supplementary Table 2). Each of the qPCR assays included 3 μL of 5X HOT FIREPol® EvaGreen® qPCR Supermix (Solis BioDyne, Estonia), 0.2 μM of forward and reverse primers, 10.4 μL of nuclease-free water, and 1 μL of DNA template. The assays were carried out using a 7500 Fast Real-Time PCR system (Invitrogen ™, USA) using the following thermal profile: initial denaturation at 95 °C for 720 s followed by 40 cycles of amplification at 95 °C for 25 s, 60 °C for 45 s, 72 °C for 60 s.

The PCR assays were performed as described in Bellankimath et al.[28]. PCR was performed in 20 μl reactions with 4 μl of 5 X HOT FIREPol® MultiPlex Mix Ready to Load (containing 10 mM MgCl₂), 0.5 μL of 10 μM primers, and 1–8 ng of template DNA. Nucleic acid- and nuclease-free water was used for the negative template control. The assay utilized a Verity 96-Well Thermal Cycler with an initial 12 min at 95 °C, followed by 30 cycles of 25 s at 95 °C, 45 s at 60 °C, and 45 s at 72 °C, ending with 7 min at 72 °C.

## Sampling criterion and clinical culture for pathogen identification and antibiotic susceptibility testing

All urine samples collected in the study were leftover samples from the urology ward, gathered according to the regulations of Justus Liebig University Giessen. All the samples collected were one day old. The samples were initially screened for bacteriuria using a dipstick and urine flow cytometry on the UF-1000i automated system, as described by Fritzenwanker et al.[62]. Only those samples that tested positive for leukocytes were included in the study (both case and control). The urology ward at JLU, Giessen, functions as the tertiary referral center

for patients with cUTI in Germany. Therefore, the cUTI case cohort included samples from patients who had one or more risk factors, such as having a urinary catheter or a history of urological procedures. The control cohort samples were from patients without symptoms who presented themselves for a preventive medical examination at the urology ward and were culture-negative.

The university hospital's microbiological laboratory performed routine microbial analysis of the collected urine samples. In brief, 10 μl of urine was inoculated onto blood agar plates (CNA Blood, MacConkey, and CPSE) and incubated for 18–24 h at 37 °C in aerobic conditions. The identification of bacterial species and antibiotic susceptibility testing (AST) were performed separately using the MADI-TOF analysis (bioMérieux, Nürtingen, Germany) and VITEK-2 systems (bioMérieux, Nürtingen, Germany). The antimicrobial susceptibility test cards of the VITEK 2 AST-N432 and ASTP611/654 Test Kits (bioMérieux, SA) were used for Gram-negative and Gram-positive organisms, respectively. The test results were interpreted in accordance with EUCAST guidelines.

## Testing of the 11 methods on complicated UTI samples

Eleven methods, comprising eight HL_SAN/M_SAN combinations and three commercial kits (Fig. 1), were evaluated on a combination of 78 patient samples collected from the Department of Urology, Pediatric Urology, and Andrology at Justus-Liebig University of Giessen in Germany. The clinical evaluation of the developed methods was carried out in two steps. In the first steps, eight SAN methods (HL + NB, H + BT, M + NB, M + BT) and three commercial methods (MC5, HZ, and NB) were tested on clinical urine samples at the Institute of Medical Microbiology, Medical Microbiome - Metagenome Unit (M3U), Justus Liebig University Giessen, Giessen, Germany. Approximately 3-8 mL of each sample was collected and processed using the above-mentioned methods without prior knowledge of the pathogen type or AST results. Depending on the number of samples received, they were processed in batches of 12–24 samples.

## Determination of method performance on the same clinical samples

After step 1, M + NB and H + NB protocols were modified to include either bead beating (H + BB + NB and M + BB + NB) or enzymatic (H + LZ + NB and M + LZ + NB) lysis steps to enhance extraction efficiency for Gram-positive pathogens. In the enzymatic version (LZ) of the methodology, the samples post-depletion were treated with an enzymatic lysis buffer (20 mg lysozyme, 20 mM Tris. Cl, 2 mM sodium EDTA, and 1.2% Triton-X-100) and incubated at 37 °C for 30 min before proceeding with the rest of the DNA extraction protocol from NB. This modification increased the time taken for DNA extraction to ~75 min. In the bead-beating (BB) version, the samples post-depletion were first treated with 20 μL of proteinase K and incubated at 55 °C for 10 min. The sample was transferred to Lysis Matrix E tubes and beaten at maximum intensity on a Vortex-Genie® 2 for 5 min. The lysis tubes were centrifuged at $\geq 10,000 \times g$ for 60 s, and the supernatant was transferred to a fresh 2 mL Eppendorf tube before continuing with the remaining NB extraction steps. The extraction steps, including the BB steps, took a total time of ~60 minutes.

Due to the limited sample volume from the clinic, not all methods could be tested on each individual sample. An overview of the methods directly compared is provided in Supplementary Fig. 9. In this regard, three samples (two culture-positive and one culture-negative) were tested using non-optimized in-house (methods 8–10), an optimized method, and one commercial method (MC5), as well as with the non-optimized in-house methods (8–10), and the optimized method alone. Each of these samples was extracted using all the methods and sequenced to facilitate comparison. Subsequently, the best-performing method was selected based on a combination of host depletion, cost, and turnaround time. The best performing method

was subsequently tested on 53 samples to evaluate its performance in comparison to the results obtained in the clinical routine. The methods' turnaround time has been detailed in Fig. 5.

### Vivalytic/PCR-based confirmation of additional pathogens from mNGS that were missed by the routine testing method

In addition to the routine culture analysis, some samples with additional bacteria detected metagenomically were reanalyzed using either Vivalytic or PCR-based tests. The Vivalytic has a sensitivity of $10^4$ CFU/mL, thereby making it sensitive enough to detect UTIs[61]. Among samples 41, 59, and 70, which had *M. morganii*, *P. stuartii*, and *P. aeruginosa* detected metagenomically, retesting was conducted through Vivalytic, as described by Hartmann et al.[61]. For samples 53, 54, and 62, where *A. urinae* was identified, and samples 56, 57, and 62, where *A. schaalii* was found, and pathogen-specific primers were employed to confirm their presence.

### Nanopore sequencing of the clinical samples

Post-extraction, all the samples were quantified using a Qubit 4.0 fluorometer with either the high sensitivity (HS-Q33231) or broad range (BR-Q-33265) assay kit. The extracted samples were purified and concentrated using the Agencourt AMPure XP system (Beckman Coulter, USA). Library preparation for MinION sequencing was conducted with the Rapid Barcoding Sequencing kit SQK-RBK114.24 (Oxford Nanopore, UK) and sequenced using MinION flow cells (R10.4.1 FLO-MIN 106D, Oxford Nanopore). Samples from both rounds were multiplexed and loaded separately onto two MinION flow cells.

### Bioinformatics analysis of the sequencing data

The sequencing data were basecalled in real-time using MinKNOW software (version 6.0.11) in fast basecalling mode with Dorado (version 7.4.13). Unclassified reads were recovered using MysteryMaster[74] with default parameters. Reads were rebasecalled with Dorado (version 0.6.4), employing HAC and SUP models to compare the results. Pathogen reads were identified using BLASTn against a custom reference genome panel of 63 common uropathogens, with low complexity regions masked using BBmask (BBMap version 38.84). BLASTn subsequently identified additional bacterial reads against the NCBI Prokaryotic Reference Genomes collection (RefProk). BLASTn was run with the following parameters: -word size 28 -max target seqs 150 -evalue 0.000001. The following cutoff values were used for bacterial identification: minimum percent identity: 80%, minimum read coverage in alignment: 65%, minimum read length: 200 nt. Reads that did not align with the RefProk database were discarded to remove human reads. If pathogen genome coverage exceeded 2%, reference-based assemblies were created using minimap2[75] (version 2.22) with the "map-ont" option. Samtools[76] (version 1.13) was used to filter the alignments (options -b -F 4) and generate a consensus sequence. ARGs were identified using abricate[77] (version 1.0.1) with the NCBI[78] and the Comprehensive Antibiotic Resistance Database (CARD)[79] databases with default settings. Virulence genes were identified the same way, using abricate with VFDB[80].

### Statistical analysis

The area under the receiver operating characteristic curve (AUROC) was calculated using routine culture as the reference standard for total DNA yield and the number of bacterial cells per ml, as identified by flow cytometry. The optimal cutoff values were computed by maximizing Youden's index[81]. Positive Predictive Value (PPV) and Negative Predictive Value (NPV) were calculated after classifying the data according to the optimal cutoff. The 95% confidence intervals for AUROC and PPV/NPV were determined through bootstrapping using 10,000 steps. The statistics described were calculated using the sklearn[82] (v1.3.2) Python package.

### Scoring metrics

Accuracy scores were calculated as follows on three levels: sample level, pathogen level, and antibiotic level.

$$\text{ACCURACY} = \frac{TP + TN}{TP + FP + TN + FN} \tag{1}$$

On sample level, true positives (TP) are considered samples flagged as positive by both routine testing and mNGS. Similarly, true negative (TN) samples are those flagged as negative by both routine testing and mNGS. False positives (FP) are samples found to be negative by routine testing that mNGS considered positive, and false negatives (FN) are samples found to be positive by routine testing but negative by mNGS. On the pathogen level, TPs are pathogens detected by both routine testing and mNGS, FPs are pathogens detected by mNGS but could not be found by routine methods, FNs are pathogens identified by routine testing that were missed by mNGS. Samples considered negative by routine testing in which no pathogen was detected through mNGS were counted as a single TN pathogen. This definition of true negative (TN) pathogens may be unintuitive and thus the sensitivity score was also reported. The recall score does not take TN and FP into account and is calculated as follows:

$$\text{SENSITIVITY} = \frac{TP}{TP + FN} \tag{2}$$

Accuracy scores on antibiotic level were determined for individual pathogen-antibiotic combinations. Any pathogen-antibiotic combination identified as resistant (R) by both routine AST and mNGS was considered TP. Similarly, pathogen-antibiotic combinations identified as susceptible (S) by both routine and mNGS testing were considered TN. FPs were those pathogen-antibiotic combinations where mNGS implied resistance while routine testing showed sensitivity, and FNs were those where mNGS implied sensitivity while routine testing showed resistance. In case of AST, specificity is particularly relevant as it shows the proportion of sensitive correctly identified as such (i.e. no overprediction of resistance), and is calculated as follows:

$$\text{SPECIFICITY} = \frac{TN}{TN + FP} \tag{3}$$

### Ethics

The study received ethical approval from the Ethics Committee of Justus Liebig University Giessen, Faculty of Medicine (AZ 158/20) for patients with UTIs and from the regional committees for medical and health research ethics (REC Helse Sør-Øst) (691177) for the healthy donors. The study had a waiver of consent for the collection and use of these samples.

### Reporting summary

Further information on research design is available in the Nature Portfolio Reporting Summary linked to this article.

## Data availability

Source data are provided with this paper. The sequencing data generated in this study have been deposited in the European Nucleotide Archive database under accession code PRJEB83412.

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

## Acknowledgements
A.B.B., S.B., J.A. and R.A. were supported by the Research Council of Norway through the projects OH-AMR-Diag (project number 336420) and UTI-Diag (project number 352514).

## Author contributions
R.A., F.W., T.H., C.I. and T.B.J. planned the study. A.B.B., S.B., J.A., and R.A. designed the experiments. A.B.B., I.K., J.A. and F.A. performed the experimental work in discussions with R.A. A.B.B., S.B. and R.A. wrote the manuscript. All authors contributed to the article and approved the submitted version.

## Funding

## Competing interests
The authors declare no competing interests.
