## [Transparent Peer Review file · Nature Communications]

Metagenomic sequencing enables accurate pathogen and antimicrobial susceptibility profiling in cUTIs in approximately four hours

Corresponding Author: Professor Rafi Ahmad

Version 0:

Reviewer comments:

Reviewer #1

(Remarks to the Author)

Reviewer summary: The authors present a systemic analysis of eleven culture independent, Nanopore sequencing-based metagenomic sequencing approaches for the detection of bacterial pathogens and antimicrobial resistance genes in urine specimens collected from individuals with complicated urinary tract infection. An optimized DNA extraction method is identified that the authors claim performs with 90% accuracy and has a turnaround time (including DNA extraction, library preparation, sequencing and basecalling) of four hours. Strengths of the article include the evaluation of multiple DNA extraction methods, direct comparison to culture and AST results in matched samples, and the use of clinical urine samples for testing and validation. Despite these strengths there are several areas of concern in the manuscript that makes it difficult to determine if the results of the experiments presented sufficiently support the conclusions made by the authors. These concerns are described in detail below:

Major concerns

1. There are statements made in the results section that do not seem to be accompanied by supporting data (or at least a text reference to supporting data). This includes line 117, lines 156-157 and lines 208-209.
2. In many instances the authors are too non-specific in their reporting. Please report quantitative measurements, effect sizes, etc instead of making general statements. An example of this is in line 117 where the authors state "The results from the nanopore sequencing of the extracted DNA indicated very similar numbers". The authors should instead explicitly state these numbers so that the reader can decide for themselves if they are similar.
3. A limitation of this work is that there is no true control cohort. It seems even though some samples were culture/mNGS negative, they were all suspected complicated UTI cases? This limitation should at least be addressed in the discussion.
4. It's a bit difficult/problematic to compare the performance of different kits/in house methods when they were all performed on different samples (Figure 2a). Each sample could contain different pathogens of differing difficulties to lyse, for example. A more fair comparison would be evaluating each method's performance on the same sample. Why wasn't this done? I think you can say the optimized protocol performed well but I am not sure how much better than other protocols you can conclude it is without a direct comparison on the same samples.
5. It is not clear if the authors are able to estimate bacterial CFU/mL by this method. It would be clinically important to know the overall bacterial load for ASB cutoffs and to know the proportion of each identified bacterial species in the sample for polymicrobial infections. I imagine one could relatively easily calculate relative abundances from the Nanopore sequencing data, but this would add time. This information would be critical; however, for clinical decision making and antibiotic selection. I don't think it would be so feasible to estimate total bacterial CFU, which I think is an important limitation of the method that needs to be discussed.
6. The reporting of the concordance between AST and mNGS ARG data is not presented in a way that is very clinically useful. Can the authors please report the concordance for each antibiotic, focusing on those that are clinically important for the management of complicated UTI? A graphical visualization of this would be very useful.

7. The logic presented for not including the CFU/mL in the AUROC calculations (line 232) makes no sense. This should absolutely be evaluated because it is important that a clinically useful culture-independent diagnostic method would accurately reflect sample CFU (flow cytometry) and species proportional abundance (mNGS).

8. The text in nearly all the figures is completely illegible at scale, please increase font size on all panels of the figures to something that is legible at 100% scale.

Moderate concerns:

1. In many sections there are statements made without supporting citations. Some of examples of these include lines 60, 62, 64, 71, 72, 75 in the introduction, as well as lines 99 and 100 in the results, among others. Please carefully go through the manuscript and make sure all statements about previous literature or statements of fact in general are supported by an appropriate citation.

2. The authors highlight throughout the article the importance of host depletion stating in urine that the host cells vastly outnumber the bacterial cells (e.g Line 85). While this may be true in the case of a healthy individual (but not always especially in healthy women), this is not usually the case in the majority of UTI scenarios where the bacterial count in urine is above 10^5 /mL. Usually when performing metagenomics on infected urine, the host contamination is a minimal issue and in these scenarios host sequences can account for as little as 5-10%.

3. It is not mentioned if the urine utilized for spike-in studies was sterile filtered? If there was an underlying microbiome this may skew the results?

4. I could not find any clinical information about the patients providing the urine samples. This includes if they were from males or females, which matters because the urinary microbiota that complicates mNGS diagnostic approaches differs by sex. Also, what type of complicated UTI cases were these? Were they all catheter-associated UTI? Please at least provide an inclusion criteria for complicated UTI here.

5. Please introduce all acronyms and abbreviations before their first use. This includes the abbreviations for the kits/methods used. Although they may be fully spelled out in a figure legend, it is not clear to the reader what they refer to exactly based on the manuscript text.

6. Were the mean accuracy scores for pathogen identification calculated on a per patient/sample (out of 50) or per pathogen basis (out of 75)? Please be more clear in your description of these results.

7. In what clinical literature is the clinical definition of a UTI 500 cells/mL. This does not seem parse with clinical guidelines and there is no reference provided.

8. Please include a separate methods section for final optimized workflow complete with timing for each step.

9. Line 350 – claims the clinical definition of UTI is 10^3 CFU/mL which I believe is incorrect by most guidelines and there is no supporting reference.

Minor concerns:

1. Line 44: “cheaper” word choice, perhaps inexpensive would be better

2. For the reference discussed in line 73, please be more specific and state the amount of concordance the previous study found and not simply that it was concordant.

3. Line 115: please replace the parenthetical with a phrase. I am also not sure how important it is because this is urine spiked with *E. coli* and not a clinical sample – if the urine was sterile-filtered before spiking then it wouldn't have host cells to deplete?

4. Line 131: can the authors be more specific about qPCR methods to measure host depletion, adding an additional descriptive term besides just “qPCR” will help the readers more automatically understand how host DNA levels were determined. Also was the host depletion analysis described in the section starting on line 128 performed on the same sample as the analysis described in the preceding paragraph?

5. Line 160: I thought the NEB Blood and Tissue kit included in its instructions a suggested bead beating step for lysing Gram positive bacteria?

6. Please make sure to spell out genus at first use and in figures unless it was previously used in the text or the abbreviation is explained in the legend.

7. I would use caution when discussing *Actinotignum schaalii* – there is still some debate about if this is an opportunistic pathogen, a commensal (especially in males), or both. This is a limitation of mNGS as it will identify species that could likely just be commensals but the field has not come to a consensus about them. Especially in a mixed sample with known

pathogens, how clinical useful would it be to know this organism was there?

8. Line 186: the authors refer to kdpDE as an identified virulence factor but do not explain what it is or why they think its important that it was identified (either in results or in discussion).

Reviewer #2

(Remarks to the Author)

This manuscript evaluates a series of DNA processing and host DNA depletion methods to enable metagenomic diagnostics for urinary tract infections. The methods were refined using spiked urine samples and then applied to clinical diagnostic samples. The novel method was assessed based on pathogen identification accuracy, antimicrobial susceptibility prediction accuracy, turnaround time, and cost. Overall, the manuscript appears to be well executed, and the writing is very clear. Although I have no concerns regarding the clarity or rigor of the work, I do not believe it meets the bar for publication in this journal for two reasons:

Novelty. The use of metagenomic sequencing for UTI diagnostics is not novel. A brief review of the literature identified four highly relevant manuscripts on this topic, as well as a recent meta-analysis covering 17 studies (Table 1). Several published reports describe rapid nanopore-based UTI diagnostics with sensitivity, specificity, and turnaround time metrics that are comparable to those reported here. Thus, the detection method, clinical application, performance metrics, and clinical context presented in this manuscript appear to have been published in reputable journals. The authors must highlight their novel contributions—presumably a refinement in sample preparation workflows—which, as currently presented, are not sufficiently novel to warrant publication in this journal.

Benchmarking. The manuscript lacks sufficient contextual benchmarking to interpret the performance metrics. Pathogen identification and antimicrobial susceptibility testing results are judged solely based on metagenomic sequencing outputs, but no comparison is provided to traditional clinical methods. Reference data from urine culture, pathogen identification by MALDI-TOF, and AST profiles via microbroth dilution or VITEK testing should be included in the main manuscript (with methodological descriptions provided) to allow the reader to understand how the metagenomic results align with established clinical data.

For these reasons, I cannot recommend this manuscript for publication in Nature Communications at this time. That said, I do encourage the authors to submit these data to a specialized clinical microbiology journal. I have posted some additional thoughts below for the authors to consider as they revise their manuscript for additional submissions.

1. Please verify the epidemiological claims in the introduction of the manuscript, particularly the claim that 40% of hospital-acquired UTIs progress to severe conditions. I am unfamiliar with this statistic, and it seems higher than expected.
2. The authors describe the use of metagenomic next-generation sequencing on clinical urine samples. Could the authors more clearly articulate the gap in the literature that this study fills? How does this study differ from previous metagenomic sequencing strategies applied to urine? How do the methods proposed here differ from those already described in the literature?
3. I applaud the use of spiked urine samples for technical validation. However, I am concerned about the use of a single strain from the various species. A panel of clinical isolates should be used, at least in the final method validation, to ensure that performance reflects a species average rather than optimization for a particular strain. This may be particularly relevant for organisms like *Proteus mirabilis* and *Pseudomonas aeruginosa*, which can have highly divergent morphotypes (e.g., mucoid variants) that could impact assay performance.
4. The authors make cost claims, but do not present sufficient supporting information. A detailed line-item list contributing to cost of goods should be included if such claims are made. Personally, I do not think that cost arguments are necessary for a scientific publication if the methods are robust and address the key problem—rapid and accurate detection of UTI pathogens and their AST profiles. I was left unconvinced by the cost computation and feel that it is largely unnecessary.
5. The discussion section could use better focus. What precisely did the authors find that addresses a gap in the literature? While multiple methods were evaluated and one was optimized, are these results different from what would be expected?
6. I found the very high number of polymicrobial infections atypical, although it may be understandable given the presumably inpatient complicated UTI cohort. Nonetheless, this point warrants better discussion and the clinical characteristics of your patient cohort should be described.
7. The authors must provide a human ethics protocol number and describe patient recruitment procedures in the Methods section.
8. I was hoping to find more robust benchmarking against established clinical methods. For example, colony counts, microbial identification by MALDI-TOF, and AST profiles by VITEK or broth microdilution could be used as references. Given that these data came from a clinical laboratory, I presume that this information could be made available.
9. Similarly, the high rate of polymicrobial infections could be cross-validated with clinical microbiology reference data.
10. Portions of the discussion highlight findings that are unsurprising, such as the prevalence of antimicrobial resistance in UTI pathogens, the efficacy of host genomic depletion, and utility arguments related to cost and screening.
11. Overall, the discussion covers a broad a scope relative to the amount of novel material presented. I believe the manuscript would be more effective as a short communication focused specifically on the optimized workflow and its advantages relative to well-established metagenomic sequencing methods for UTIs.

Figures:

- Figures 2A and 2B might be better presented as a table or confusion matrix. It is not clear what the expected versus

observed outcomes are, which makes it difficult to assess method efficacy.

- Figure 2C lacks a gold standard reference. It is unclear whether species co-occurrence is based on novel metagenomic findings or a traditional culture-based system. This distinction is important.
- Similar concerns apply to Figures 3A, 3B, and 3C. I would prefer to see a comparison of expected versus observed antibiotic susceptibility results.
- The ROC curves shown in Figure 4 are clear. However, the concept of DNA screening feels disconnected from the manuscript's main narrative.

Reviewer #3

(Remarks to the Author)

This study developed a rapid, selective, and cost-effective host depletion-based metagenomic workflow for analyzing UTI samples and compared it with existing detection workflows to demonstrate its theoretical advantages. The study involved substantial workload and has practical significance; however, the experimental design is not complex and lacks originality. In addition, I have the following concerns:

1. Line 471: The concentration of human cells/nucleic acids is an important factor affecting the sensitivity of mNGS detection. The authors used WBCs from the human buffy coat in the "spiked urine samples," but did not mention the number or concentration of these cells. This is inappropriate. Please provide this information. What level of human nucleic acid background was used when designing the spiked urine samples and conducting the experiments? In other words, it is unclear whether the "spiked urine samples" had high or low background. A more advanced experimental design would involve preparing quality control samples with varying concentrations of human nucleic acids to evaluate the method's sensitivity under different background conditions. This would better reflect clinical scenarios, where urine samples from patients with UTIs show great variability in leukocyte content. The authors are encouraged to consider this.
2. Line 545–546: Please explain in detail the detection range of qPCR and Vivalytic tests.
3. Line 574–575: Please specify the positivity threshold for ARGs. This was overlooked. What parameters did the authors use to define positive versus negative detection?
4. Figures 2c and 3c: These figures are not the most effective ways (figures) to illustrate co-detection of bacteria and resistance genes. Moreover, the figures lack values showing the frequency or proportion of each combination, which is confusing. Consider replacing them with a clearer visualization. Also, for Figure 2 and Supplementary Figure 6, please display the actual values on top of each bar.
5. Line 105: "patent" should be "patient."?
6. Line 186: The authors mentioned the detection of virulence genes here, but the Methods section does not describe how virulence genes were detected or reported. Please add this information.
7. Legend of Supplementary Figure 3: "10⁵ CFU/mL" should be written as "10⁵ CFU/mL."
8. Line 204–205: Why was this statement made? Is there experimental data supporting it, or is it just empirical (which is not appropriate in a scientific paper)? Also, how are "a low pathogen load" and "very high leukocyte load" defined? The current study design does not provide a basis for drawing such conclusions.
9. A general issue throughout the manuscript is that many of the numerical values presented are not supported by raw data or calculation methods. For instance, we do not know how the values in Line 201 (17%), Line 203 (19%), Line 207 (170%), and Line 208 (1%) were derived. Even in Supplementary Figure 7, there are bare plots without any numeric annotations. This appears highly unrigorous.
10. Supplementary Figure 8: Please clearly display both values and percentages above each bar.
11. Line 223–244: Due to various influencing factors during experimentation, DNA yield may show significant inter-batch variation. Furthermore, the complexity of urine varies among patients depending on disease severity. Therefore, using a small sample size to establish a DNA yield threshold for screening is prone to significant error and may not be generalisable to larger or different populations. If the authors intend to demonstrate that this analysis is meaningful or propose DNA yield as a screening or diagnostic marker, they should validate it using an independent cohort. Otherwise, this result should not be highlighted in the main text or abstract.
12. Line 249: It is recommended to present both percentages and absolute values, e.g., "90% (45/50)," to aid understanding.
13. Line 252: Replace "the selected method" with "the optimized method."
14. Line 252–254: Turnaround time (TAT) usually depends on the number of samples processed per batch. There is no information in the manuscript about how many samples can be tested in one batch. How many samples can be processed on one chip? For clinical application, can the workflow support sample-on-demand testing?
15. Line 257: Does the mentioned cost include the reagents used for host depletion?

Version 1:

Reviewer comments:

Reviewer #1

(Remarks to the Author)

1. Figure 1: Protocol details are missing and need to be clearly presented to support the idea that it could be completed within 4 hours.
 - a. For the step where you add 10uL of HL-SAN/M_SAN and incubate at 37C – what is the duration of the incubation?
 - b. How long does the pellet 2x wash step take? Account for resuspension and centrifuge time.
 - c. How long was the treatment with the enzymatic lysis buffer for methods 6-9 and how long does the treatment with proteinase K and bead beating take for methods 10 and 11?

- d. Please provide the exact duration of the “subsequent extraction steps” for methods 6-11.
- e. Please provide a time estimate for the concentration and purification by Ampure XP.
- f. You are missing a quantitation step before starting library preparation. Nanopore suggests doing this by qubit. Please include your quantitation method and the duration of doing this for 12-24 samples (as this is noted as the most common sample size in your methods)
- g. The steps of flow cell priming and loading are not included – please include these as well as the time that must be allotted for these steps.

2. Cost Analysis: There seems to be some inaccuracy in the cost analysis presented in lines 328-335. For the sequencing costs, a single minION flow cell is \$800 and the rapid barcoding kit for 24 samples is also \$800. With these numbers, the price per sample would be ~\$67/sample not include any third party reagents and consumables needed for the full procedure. Also I believe there is a typo in line \$335 – it is stated that DNA extraction costs approximately \$6, which I believe should be \$36. Also this cost estimate assumes multiplexing of 24 samples, but wouldn't having to wait for 24 samples to be ready to test affect turnaround time? Even in a high volume practice, it would likely take at least an hour or two for this many samples to be submitted for testing. This needs to be either factored into the turnaround time, cost, or discussed as a limitation.

3. Given there is no formal analysis of the method to distinguish true case and controls, the claim that the method can “diagnose UTI” cannot be made. UTI diagnosis requires symptoms (i.e. a host immune response) and the presence of causative pathogen. As it stands, it would be more accurate to say this this method can detect bacteriuria in individuals with suspected complicated UTI. The AUROC analysis presented in figure 4 is only assessing the accuracy in discerning culture positive from culture negative. Although the authors have added to the methods section that a control group of samples were collected from patients without symptoms, it is unclear if these all perfectly corresponded to the culture negative samples (or if some were culture positive). There needs to be a true case/control analytical design (ideally of the mNGS prediction) to claim that the methods performs UTI diagnosis. It's also a bit confusing because in the sentence just before it says “only those samples that tested positive for leukocytes and nitrate were included in the analysis.” Does this include the control samples?

4. Frankly, the authors response to my question about CFU/mL estimation does not makes sense. Colony forming unit assays are the gold standard for determining the absolute abundance of a given bacterial pathogen in a clinical sample. It is NOT measured after incubation. The urine is plated directly onto the agar and each colony that arises is representative of a single bacterium that founded it. It is exactly an estimation of the number of bacteria in the sample at the time of plating. The issue with current clinical urine culture methods is not that CFU/mL counts of pathogens are not accurate, its that they can fail to capture more fastidious pathogens that need specialized conditions to grow. You state correctly that mNGS alone only gives relative abundance, so if it needs to be coupled with flow cytometry to gain an accurate estimate of absolute abundance, the time for this additional method should be added to the overall turnaround time if the goal is more than just pathogen detection.

5. To clearly show benchmarking of antimicrobial susceptibility in a clinically useful way, the authors need to include Supplemental Figure 8 in the main text. Showing the performance of mNGS for AST prediction by antibiotic and pathogen is exactly the point of the analysis you are showing and is more useful to the reader than Figure 3d. Co-occurrence of antibiotic resistance in samples is not the main point of this analysis and this sub-figure should be moved to the supplement. The figures in Supp Fig 8 should replace it in the main text figure 3.

6. The title should be changed to more accurately represent the findings of the study based on my comments above that may lead to a revision of the actual TAT and a needed clarification that the mNGS method alone has been shown to perform pathogen and/or bacteriuria detection but not formally UTI diagnosis.

Reviewer #3

(Remarks to the Author)

Thanks to the authors for addressing all my comments

Version 2:

Reviewer comments:

Reviewer #1

(Remarks to the Author)

1. I think four hours is a very optimistic estimate and do not fully believe that both pathogen ID and antimicrobial gene profiling will reproducibly be achieved in this time frame. I suggest the addition of "approximately" to the title and other sentences where the four hour time window is discussed.

2. Antimicrobial resistance gene identification is not the same as antimicrobial susceptibility testing. The use of "antibiotic susceptibility" in the title makes the reader believe that AST can be done by this method when what is actually being done is AMR gene profiling. Even if the predictions of from the AMR gene profiling are relatively concordant with AST, "antibiotic susceptibility profiling" should not be used to describe the detection of AMR genes due to its connotation. I suggest

replacing "susceptibility" with "gene" in the title to avoid confusion.

Reviewers' comments:

We thank the reviewers for their valuable comments for improving the quality of the manuscript. Below is the point-by-point response to the comments raised by the reviewers:

Reviewer #1:

Major comments

1. There are statements made in the results section that do not seem to be accompanied by supporting data (or at least a text reference to supporting data). This includes line 117, lines 156-157 and lines 208-209.

We thank the reviewer's comment. Relevant sections of the manuscript have been revised to include the calculations and supporting data throughout.

2. In many instances the authors are too non-specific in their reporting. Please report quantitative measurements, effect sizes, etc instead of making general statements. An example of this in is line 117 where the authors state "The results from the nanopore sequencing of the extracted DNA indicated very similar numbers". The authors should instead explicitly state these numbers so that the reader can decide for themselves if they are similar.

We thank the reviewer for this review. The revised manuscript has been modified to make it more specific in reporting by including relevant calculations and numbers.

3. A limitation of this work is that there is no true control cohort. It seems even though some samples were culture/mNGS negative, they were all suspected complicated UTI cases? This limitation should at least be addressed in the discussion.

We thank the reviewer for this point. We would like to clarify that the control group consisted of patients without symptoms who presented themselves for a preventive medical examination at the urology ward. We have revised the Methods section to clarify this point (Lines 620-624).

4. It's a bit difficult/problematic to compare the performance of different kits/in house methods when they were all performed on different samples (Figure 2a). Each sample could contain different pathogens of differing difficulties to lyse, for example. A more fair comparison would be evaluating each method's performance on the same sample. Why wasn't this done? I think you can say the optimized protocol performed well but I am not sure how much better than other protocols you can conclude it is without a direct comparison on the same samples.

We thank the reviewer for this valuable comment. We agree that direct comparison is important, and we have performed this analysis; however, it was not well presented in the manuscript. We have now updated the manuscript to clarify this point. The results (Lines 247-273), figures (2 & 3), supplementary material (Supplementary Figures 9 and 10), and methods (Lines 650-670) have been updated in the revised version.

5. It is not clear if the authors are able to estimate bacterial CFU/mL by this method. It would be clinically important to know the overall bacterial load for ASB cutoffs and to know the proportion of each identified bacterial species in the sample for polymicrobial infections. I imagine one could relatively easily calculate relative abundances from the Nanopore sequencing data, but this would add time. This information would be critical; however, for clinical decision making and antibiotic selection. I don't think it would be so feasible to estimate total bacterial CFU, which I think is an important limitation of the method that needs to be discussed.

We thank the reviewer for the insightful comment. We agree with the reviewer that bacterial load is crucial for distinguishing asymptomatic bacteriuria. We already obtain read numbers per pathogen (see Supplementary Table 2), which does not require additional analysis time. As discussed, CFU/mL is measured after incubation, while mNGS reads reflect bacterial abundance in the sample at the time of DNA isolation, without culture. That's why the read abundances were compared to bacterial/host numbers from flow cytometry, as these results come from the same culture-free sample quantities, not CFU/mL. The CFU/mL for each sample was calculated during routine clinical processing and is shown in Supplementary Table 3. Furthermore, the detection limit for the optimized method corresponds to 10^3 CFU/mL.

6. The reporting of the concordance between AST and mNGS ARG data is not presented in a way that is very clinically useful. Can the authors please report the concordance for each antibiotic, focusing on those that are clinically important for the management of complicated UTI? A graphical visualization of this would be very useful.

We thank the reviewer for this comment. This information is available in Supplementary Figures 7 and 8, where we visualized the concordance stratified in several ways in addition to the per-sample concordance presented in Figure 3; by antibiotic, by species, and for each pathogen-antibiotic combination, annotated with the relevant resistance mechanisms, etc. All the included antibiotics are based on the routine ATS panel (VITEK-2 AST-N432 and ASTP611/654) and include antibiotics that are clinically relevant for UTI treatment.

7. The logic presented for not including the CFU/mL in the AUROC calculations (line 232) makes no sense. This should absolutely be evaluated because it is important that a clinically useful culture-independent diagnostic method would accurately reflect sample CFU (flow cytometry) and species proportional abundance (mNGS).

We thank the reviewer for bringing this point to our attention. CFU/mL is used for determining culture positivity in routine testing. This was used as the outcome variable in AUROC calculations. To clarify this, we have now included an AUROC analysis of CFU/mL in Supplementary Figure 14 (Line 313). We also agree with the second point, and this information is available in Supplementary Figure 12 and presented in the manuscript (Lines 287-294), where we have investigated how bacterial/host abundance obtained through flow cytometry correlates with those obtained through mNGS (read abundance).

8. The text in nearly all the figures is completely illegible at scale, please increase font size on all panels of the figures to something that is legible at 100% scale.

The images have been updated in the revised manuscript.

Moderate concerns:

1. In many sections there are statements made without supporting citations. Some of examples of these include lines 60, 62, 64, 71, 72, 75 in the introduction, as well as lines 99 and 100 in the results, among others. Please carefully go through the manuscript and make sure all statements about previous literature or statements of fact in general are supported by an appropriate citation.

We thank the reviewer for their comment on supporting citations. Appropriate citations/supporting text have been included in the introduction section in the revised version of the manuscript.

2. The authors highlight throughout the article the importance of host depletion stating in urine that the host cells vastly outnumber the bacterial cells (e.g Line 85). While this may be true in the case of a healthy individual (but not always especially in healthy women), this is not usually the case in the majority of UTI scenarios where the bacterial count in urine is above 10^5 /mL. Usually when performing metagenomics on infected urine, the host contamination is a minimal issue and in these scenarios host sequences can account for as little as 5-10%.

We thank the reviewer for this comment. From the flow cytometry data included in the study (Supplementary Table 4), we observe a variable immune response among the cUTI patient samples, resulting in different levels of background host cells (10^1 - 10^5 /mL). Additionally, DNA extraction tends to favor host cells over bacterial cells due to their larger genome sizes. Host cells, such as leukocytes and epithelial cells, have genomes of about 6 billion bases, whereas bacterial genomes average around 5 million bases. Consequently, host cells contribute disproportionately high amounts of DNA, so even a small number of host cells can yield a large DNA quantity. Moreover, the structural properties of eukaryotic cells make them easier to lyse compared to bacterial cells. As a result, even with a high number of bacterial cells, the extracted DNA may still be dominated by host DNA. Furthermore, we performed sequencing on two clinical samples (sample 14 and 16) using the NB method, which does not include a host depletion step. The bacterial counts were 10^4 and 10^5 CFU/mL, respectively, while the host cell counts were 10^3 CFU/mL for both samples. The resulting data show that 96-99% of reads are host reads. The sequencing data and flow cytometry results can be found in supplementary tables 4 and 2.

3. It is not mentioned if the urine utilized for spike-in studies was sterile filtered? If there was an underlying microbiome this may skew the results?

We thank the reviewer for the comment on the underlying microbiome. The spiked urine samples were not filtered because this might remove any background host cells. To examine any underlying microbiome, we cultured urine samples from all batches used for spiked samples

to check for its presence. Additionally, to verify this, we also did not detect any microbiome from the sequencing data generated from the spiked samples shown in Supplementary Table 2 and Supplementary Figure 1.

4. I could not find any clinical information about the patients providing the urine samples. This includes if they were from males or females, which matters because the urinary microbiota that complicates mNGS diagnostic approaches differs by sex. Also, what type of complicated UTI cases were these? Were they all catheter-associated UTI? Please at least provide an inclusion criteria for complicated UTI here.

Thank you for this insightful comment. Indeed, we agree that the urinary microbiota can vary by sex and influence the mNGS-based approach. We have included gender-specific information in the results section of the revised manuscript and provided a more detailed account in Supplementary Table 3. Not all samples were catheter-associated. This information has also been added to Supplementary Table 3. Additionally, we amended the methods section to further describe the inclusion criteria (Lines 620-624). However, our ethical approval prevents us from sharing any further clinical or patient details beyond those already provided.

5. Please introduce all acronyms and abbreviations before their first use. This includes the abbreviations for the kits/methods used. Although they may be fully spelled out in a figure legend, it is not clear to the reader what they refer to exactly based on the manuscript text.

The correction has been made in the revised version of the manuscript.

6. Were the mean accuracy scores for pathogen identification calculated on a per patient/sample (out of 50) or per pathogen basis (out of 75)? Please be more clear in your description of these results.

We thank the reviewer for this valuable suggestion. Calculations were performed at different levels: sample, pathogen, and antibiotic. We have included a table (Table 1) to provide a clear overview and have clarified the descriptions of the results throughout the manuscript. Additionally, we have added a detailed section in the methods that outlines exactly how these metrics were calculated (Lines 713-734).

7. In what clinical literature is the clinical definition of a UTI 500 cells/mL. This does not seem parse with clinical guidelines and there is no reference provided.'

We appreciate the reviewer for bringing this to our attention. The statement has been taken out of the revised manuscript.

8. Please include a separate methods section for final optimized workflow complete with timing for each step.

We thank the reviewer for this suggestion. We respectfully refer to Figure 5 in the manuscript, which outlines the steps of the optimized method along with their respective timings. Additionally, Supplementary Figure 14 offers a comparative overview of the optimized method and tested commercial methods.

9. Line 350 – claims the clinical definition of UTI is 10^3 CFU/mL which I believe is incorrect by most guidelines and there is no supporting reference.

The statement has been removed from the revised version of the manuscript.

Minor concerns:

1. Line 44: “cheaper” word choice, perhaps inexpensive would be better

The correction has been made in the revised version of the manuscript.

2. For the reference discussed in line 73, please be more specific and state the amount of concordance the previous study found and not simply that it was concordant.

The introduction section of the manuscript has been revised to better highlight the existing gap in the literature.

3. Line 115: please replace the parenthetical with a phrase. I am also not sure how important it is because this is urine spiked with *E. coli* and not a clinical sample – if the urine was sterile-filtered before spiking, then it wouldn't have host cells to deplete?

The urine samples were not sterile filtered, but they were cultured to check for sterility. Additionally, the sequencing data (Supplementary Table 2) do not indicate any background contamination from the samples. Please see our response to point 3 under moderate concerns.

4. Line 131: can the authors be more specific about qPCR methods to measure host depletion, adding an additional descriptive term besides just “qPCR” will help the readers more automatically understand how host DNA levels were determined. Also was the host depletion analysis described in the section starting on line 128 performed on the same sample as the analysis described in the preceding paragraph?

We thank the reviewer for this suggestion. Additional information has been added to the methods section of the revised manuscript. Separate batches of urine were used for comparing the in-house methods, the endonuclease experiment, comparing the optimized method against commercial methods, and testing the optimized method on a panel of uropathogenic sequence types. These four sets of experiments were performed with four different batches of urine, evaluated using dipstick tests, and for the background microbiome by culturing.

5. Line 160: I thought the NEB Blood and Tissue kit included in its instructions a suggested bead beating step for lysing Gram-positive bacteria?

The blood and tissue used in the current study are from QIAGEN (Line 549) “DNasey Blood and Tissue kit (cat: 69504, Qiagen, Germany).” The kit was used following the manufacturer's instructions and does not include a bead-beating step. The manufacturer recommends using an enzymatic lysis buffer for gram-positive bacteria, which must be purchased separately and is not provided as part of the kit.

6. Please make sure to spell out genus at first use and in figures unless it was previously used in the text or the abbreviation is explained in the legend.

We thank the reviewer for this suggestion and agree that it would be helpful to include the full names. However, when we tried to include the full names in the figures, they became completely illegible, and the word limit on figure legends prevents us from including the full names there. Therefore, we have not included the full names in the figures. Instead, the complete details about pathogen identification, including full names, are provided in the supporting materials (Supplementary Table 2), and a line referring to this has been added to the Figure 2 legend.

7. I would use caution when discussing *Actinotignum schaalii* – there is still some debate about if this is an opportunistic pathogen, a commensal (especially in males), or both. This is a limitation of mNGS as it will identify species that could likely just be commensals but the field has not come to a consensus about them. Especially in a mixed sample with known pathogens, how clinically useful would it be to know this organism was there?

We thank the reviewer for this suggestion. Indeed, there has been a lack of consensus on the status of *Actinotignum schaalii*, which led us to take further steps and confirm the presence of this species using additional molecular methods such as PCR or Vivalytic. We report this species here because, ultimately, it is the clinician who decides the best course of treatment for the patient, based on all available information. The relevant points have been added to the discussion section (Lines 402-409) in the revised manuscript.

8. Line 186: the authors refer to kdpDE as an identified virulence factor but do not explain what it is or why they think its important that it was identified (either in results or in discussion).

The manuscript discussion part has been modified to reflect the point (Lines 449-452)

Reviewer #2:

Novelty of the study: We appreciate the reviewer's comments and would like to clarify the novelty of our work. Although metagenomics has been used for UTI diagnosis, our manuscript presents unique novel features and clinical utility that distinguish it from existing reports. The following points highlight the innovation, appropriate benchmarking, and demonstration of clinical utility of our metagenomic method.

- **Benchmarking of the metagenomics results:**

- The results have been systematically benchmarked against clinical data obtained from the routine microbiology laboratory. This methodology serves as the gold standard and represents common practice when comparing the mNGS approach.
- The pathogen identification results have been benchmarked against the MALDI-TOF data.
- AST predictions have been benchmarked against VITEK-2 data, which was performed using AST-N432 and ASTP611/654 Test Kits.

- The complete clinical data are included in Supplementary Table 1.
- The methodology includes details on the MALDI-TOF and VITEK-2 testing procedures performed in the routine laboratory.
- **Comparison of different methods on the same sample:** A part of the study also involved testing multiple methods on the same sample. While comparing all 11 methods on the same samples is challenging due to practical limitations on the amount of patient urine available, we have compared a set of methods (four in-house and one commercial) on the same samples. In this regard, five samples (three culture-positive and two culture-negative) were tested using four of our in-house methods and one commercial method (MC5). Each of these five samples was extracted using all five methods and sequenced to facilitate comparison. We have also compared additional samples using different methods and can include the data in the revised version of the manuscript. To further validate the performance of the optimized method, we are currently testing it on over 250 clinical urine samples.
- **Demonstration of clinical utility:**
 - **Rapid turnaround time:** We report a TAT of 4 hours, which is less than half of what has been previously reported, 10 hours by Liu et al., 2023. This is of significant clinical importance for both UTI management and prudent antibiotic use.
 - **High Accuracy for both UTI positivity, pathogen detection and AST:** We have reported 100% UTI positivity, 99% pathogen identification, and 91% AST detection rates. This is currently unmatched by any published studies. Most studies on AST are either limited to only selected antibiotic classes. We have presented our AST data for 34 unique antibiotics, covering the complete panel of antibiotics used in the urology ward at Giessen.
 - **Cost:** The in-house methods developed and tested are cost-effective and particularly well-suited for a low-resource setting. To our knowledge, this is the only study on UTI mNGS that reports the cost aspects of its method or compares it to commercial methods.
 - **Flow cytometry and DNA yield as parameters for culture positivity/negativity:** In this study, we performed ROC analysis on the flow cytometry and the DNA yield using the routine culture as a reference. The primary goal was to utilize flow and DNA yield data to predict whether a sample would be culture-positive or negative rapidly when clinical data is unavailable or not yet ready. Moreover, we included the DNA yield in the AUROC analysis, as determining cell counts from a patient sample is not always part of standard operating procedures in all testing centres/hospitals. Hence, this can be used as a pre-screening parameter to select samples for sequencing, thereby reducing overall costs.

These are entirely novel results that have not been reported in any prior metagenomic next-generation sequencing (mNGS) studies.

- **Additional novel aspects points about the study:**
 - Latest ONT chemistry: This is the only mNGS study on patient UTI samples that has utilized the new 10.4 chemistry. All previously published studies have been conducted with the old and discontinued 9.4 chemistry.
 - Sample types: The current study has investigated complex urinary tract infections, whereas the listed studies have primarily focused on uncomplicated urinary tract infections (UTIs), with only a few also examining complicated samples. Our method, on the other hand, achieved 99% species-level accuracy on complex UTI samples, where 46% of samples were polymicrobial and contained as many as five unique pathogens. This reflects the complexity of typical complicated UTI cases.
 - Methods: Tested and compared multiple (11) methods on clinical samples.
 - A unique aspect is the AMR predictions. None of the listed studies perform extensive AMR predictions; instead, they restrict themselves to either identifying AMR genes or correlating only with well-known classes, such as β -lactamase genes.
 - We have also examined the co-occurrence of resistance and pathogens to uncover the complexity of clinically complicated UTI samples.
 - Most of the studies listed are from the Asian context, while this study is set in a European context. This distinction makes the research unique, as different variations, such as AMR burden, population, and clinical practices, influence the samples.
 - AUROC for sample CFU: In accordance with the reviewers, we have also performed the ROC analysis for the bacterial CFU data, and the graph is attached in the following section. The analysis resulted in an AUROC of 1 and a CFU cutoff of 10^3 CFU/mL.

The table below highlights the novel points of our approach in comparison to published mNGS studies on UTIs.

Study	Summary
Liu et al., 2023	This was a retrospective study where the authors have evaluated >1000 patient samples.  • The total turnaround time was 10 hours.  ○ This is more than double the amount of time of our protocol of 4 hours. This has significant clinical significance both for UTI clinical management and prudent antibiotic usage.

- The reported method showed a sample-level positive identification rate of only 71% and an overall false positive identification rate of 12%. Furthermore, only 53% of the correctly identified positive samples matched completely. This means the overall exact matching positive samples only accounted for 38%.
 - Our optimized method achieves 100% UTI positivity rate and 99% sample-level positive prediction, missing only a single pathogen in a mixed sample where the other three pathogens were correctly identified. Our method identified several pathogens, not identified by routine culturing, however, the presence of all these pathogens was confirmed using gold-standard methods like Vivalytic, showing our method has higher sensitivity than routine culturing methods.
- The reported study did not evaluate complex UTI samples. The majority (57%) of the samples evaluated in this study were negative samples and only 4.4% were polymicrobial, containing only two pathogens.
 - Our method on the other hand, achieved 99% species level accuracy on complex UTI samples where 46% of samples were polymicrobial and contained as many as five unique pathogens. This reflects the complexity of typical complicated UTI cases.
- AST was limited to beta-lactamase genes in *E. coli* and *K. pneumoniae*. This approach fails to capture the complexity of resistance and has a limited clinical relevance. Even in this limited analysis, their method predicted resistance correctly in only 75% of resistant samples, and overpredicted resistance in 57% of susceptible samples. This could lead to problematic antibiotic overuse.
 - Our study on the other hand, evaluated the concordance between mNGS-predicted and routine AST for the entire AST panel (as used in the clinical routine), agnostic of the pathogen. We evaluated 34 unique antibiotics across 14 pathogen species, including in complex polymicrobial samples. Here, we showed >90% concordance, which has never been reported using an mNGS approach.
- Additionally, there are significant differences in methodology. This study employed HL_SAN. Finally, the DNA extraction method also varies, as the study utilized the DNeasy Blood & Tissue Kit (Qiagen, 69504) according to the manufacturer's instructions.
- We have tested 11 different DNA extraction method including several host depletion and microbial enrichment steps. While we benchmark both HL_SAN and M_SAN, using M_SAN in the optimized protocol. In contrast, we have tested multiple extraction methodologies (including the DNeasy Blood & Tissue Kit (Qiagen, 69504), with the optimized method being based on magnetic bead-based extraction.
- There are also notable differences in the library preparation steps. The study uses a PCR barcoding kit that amplifies the samples before sequencing, while we have used a Rapid barcoding kit without any amplification. Moreover, the study utilized the outdated and discontinued R9.4 Nanopore chemistry, while we have used the latest and current 10.4 chemistry, which have major differences.

Bellankimath et al., 2024	This is our own first proof-of-concept study on the direct extraction of DNA from spiked urine samples and not on patient samples.  • The study benchmarked three commercial kits without any host depletion capabilities. • It was performed only on samples spiked at 10^5 and 10^3 CFU/mL. No clinical samples were tested in this study.
Zhang et al., 2022	The authors have optimized a metagenomic nanopore sequencing (mNPS) test for pathogen detection and identification in urine samples of 76 patients with acute, uncomplicated UTIs. 20 samples were used to assess the three methods of library preparation (rapid barcoding sequencing with RBK004, 16S sequencing with RAB204/16S024, and PCR barcoding sequencing with PBK004)  • The samples used for this study were from acute uncomplicated UTI patients. ARGs have been assigned AB classes, but there is no correlation of the detected ARGs against each antibiotic. • There is no host depletion being performed in the study. The samples were centrifuged before extracting the DNA using a combination of Metapolyzyme and the Indispin pathogen kit. • The authors have used a PCR barcoding kit with modified extension time and amplification cycles. We have used a rapid barcoding kit. Moreover, the kit and the sequencing were performed on the 9.4 version of the kit and flow cells. • The authors haven't performed any correlation of the detected ARGs with the antibiotic susceptibilities. The ARGs have been assigned AB classes, but there is no correlation of the detected ARGs against each antibiotic.
Jia et al., 2023	In this study, 33 patient samples with previously negative urine culture results but still exhibiting persistent urinary tract symptoms were evaluated using mNGS. The authors utilized the QIAamp DNA Micro Kit for DNA extraction, followed by sequencing the samples with the QIAseq™ Ultralow Input Library Kit (Illumina). The study design differs significantly in terms of sample selection criteria, sample processing, and sequencing platform. No rapid TAT.
He et al., 2024	This is a systematic review of 17 studies, of which nine were prospective and eight were retrospective. All three studies mentioned above are also included in this count. Five studies used Nanopore sequencing, while the others utilized Illumina, Ion Torrent, etc. Of these five, two (Liu, 2023 and Zhang, 2022) where Nanopore was used have been mentioned above. Wang, 2023: A Nanopore-based mNGS study with the same study design as that of authors Zhang et al., 2022. The authors used the same kit for DNA extraction and employed a PCR barcoding kit for library preparation prior to sequencing. Furthermore, the study only identifies the causative pathogens, without reference to ARG detection or AST predictions. Moreover, the kit and the sequencing were performed on the 9.4 version of the kit and flow cells.

Wang, 2022: Used a commercial kit for extraction (Sansure DNA Extraction Kit). The authors employed a targeted approach where the 16S rRNA gene was amplified before sequencing on the Nanopore platform. Moreover, the kit and the sequencing were performed on the 9.4 version of the kit and flow cells.

Zhao, 2024: The study tested 297 samples distributed across seven biological sample types. Of the 297 samples, only 24 were urine samples. The authors used host depletion protocols as described by Nelson et al., 2019. The samples were amplified using a PCR barcoding kit before sequencing on the Nanopore platform. Moreover, the kit and the sequencing were performed on the 9.4 version of the kit and flow cells.

We have also compared our optimized method against the published methods: Pathogen identification concordance between the optimized method and clinical routine results (n=88, accuracy 99%, precision 100%, recall 99%) was higher than results reported in comparable studies, while turnaround time was the shortest reported at only 4 hours: Liu et al., 2023 n=1327 accuracy 95% (precision 64%, recall 88%)(Liu et al., 2023), Zhang et al., 2022 n=75 accuracy 92% (precision 98%, recall 87%)(Zhang et al., 2022), Janes et al., 2022 n=81 accuracy 89% (precision 87%, recall 79%)(Janes et al., 2022), Jia et al., 2023 n=43 accuracy 77% (precision 34%, recall 50%)(Jia et al., 2023).

In addition, we have revised the discussion and introduction sections of the manuscript to highlight its novelty and clinical relevance more effectively, and we hope this clarifies the study's uniqueness.

Benchmarking metrics used in the study:

We sincerely thank the reviewer for the comment. We respectfully note that the benchmarking methods are included in the section “**Sampling criterion and clinical culture for pathogen identification and antibiotic susceptibility testing**” within the methodology part of the manuscript. In this study, we have benchmarked all metagenomics results to MALDI-TOF data for pathogen identification and VITEK-2 data for antibiotic susceptibility. Both MALDI-TOF and VITEK-2 are regarded as the gold standard methods for pathogen identification and antibiotic susceptibility testing, respectively, in routine clinical practice. Additionally, many metagenomics studies, such as those involving UTIs, sepsis, and stool, use these methods to validate their metagenomic results.

However, it may not have been clearly emphasized or presented in the context of benchmarking in the manuscript and the figures. Therefore, the results, discussion, and figures have been modified to enhance the presentation of the benchmarking results.

A comprehensive list of studies, along with the corresponding benchmarking tool, is provided below.

Clinical studies	Sample type	Benchmarking methods for pathogen identification	Benchmarking methods for antibiotic susceptibility
Liu, M. et al. Transl Med 13, e824 (2023).	Urine	MALDI-TOF MS	VITEK 2 compact or Disc diffusion
Zhang, L. et al. Front. Microbiol. 13, 858777 (2022).	Urine	VITEK MS (MALDI-TOF)	VITEK 2 Compact Systems
Jia, K. et al. Front. Cell. Infect. Microbiol. 13, (2023).	Urine	Urine culture (Not specified)	Urine culture (Not specified)
Wang, Y. et al. J Transl Med 21, 762 (2023).	Urine	VITEK MS (MALDI-TOF)	Disc diffusion
Wang, Y. et al. Diagnostics (Basel) 12, 2128 (2022).	Urine	Urine culture (Not specified)	Urine culture (Not specified)
Janes, V. A. et al. The Lancet Microbe 3, e588 e597 (2022).	Urine	MALDI-TOF MS	VITEK-2 or Disc diffusion
Charalampous, T. et al. Genome Med 13, 182 (2021).	Sputum/respiratory sample	MALDI-TOF	Disc diffusion according to EUCAST guidelines
Wang, H. et al. PLOS ONE 15, e0232610 (2020).	Sputum/respiratory sample	VITEK MS (MALDI-TOF)	Not performed
Dong, Y. et al. Infection and Drug Resistance 16, 677–694 (2023).	Sputum/respiratory sample	MALDI-TOF	VITEK 2 Compact
Algethamy, H. et al. Sci Rep 15, 3291 (2025).	Sputum/respiratory sample	VITEK MS (MALDI-TOF)	VITEK 2
Hauser, S. et al. Microbiology Spectrum 11, e01294-23 (2023).	Sputum/respiratory sample	MALDI-TOF	Not performed
Wang, W. et al. Front. Cell. Infect. Microbiol. 14, (2024).	Sepsis	MALDI-TOF	VITEK 2 Compact
Harris, P. N. A. et al. Microbiol Spectr 12, e03065-23 (2024).	Sepsis	VITEK MS (MALDI-TOF)	VITEK 2
Zhou, M. et al., Sequencing. Front. Genet. 12, (2021)	Sepsis	MALDI-TOF	Standard broth microdilution

Lee, I.-K. et al. Journal of Microbiology, Immunology and Infection 55, 845–852 (2022).	Sepsis	VITEK MS	VITEK 2
Sun, L. et al. Front. Cell. Infect. Microbiol. 12, (2022)	Sepsis	VITEK compact	2 Not performed
Zhou, Y et al. Sci Rep 13, 9460 (2023).	Sepsis	MALDI-TOF	Not performed
Shi T et al. , J Orthop Surg Res. 2024 May 17;19(1):301.	Stool	MALDI-TOF MS	Not performed
Street, T. L. et al. J Clin Microbiol 60, e02156-21 (2022)	Periprosthetic joint	MALDI-TOF MS	BD PHOENIX 100 or Disc diffusion
Peterson, C.-L. et al. Microorganisms 10, 441 (2022).	Periprosthetic joint	MALDI-TOF MS	VITEK 2
Tan, J. et al. Front. Cell. Infect. Microbiol. 14, (2024).	Stool	VITEK Compact	2 VITEK 2 Compact

1. Please verify the epidemiological claims in the introduction of the manuscript, particularly the claim that 40% of hospital-acquired UTIs progress to severe conditions. I am unfamiliar with this statistic, and it seems higher than expected.

We thank the reviewer for this comment. The introduction section has been updated in the revised manuscript (Lines 56-58).

2. The authors describe the use of metagenomic next-generation sequencing on clinical urine samples. Could the authors more clearly articulate the gap in the literature that this study fills? How does this study differ from previous metagenomic sequencing strategies applied to urine? How do the methods proposed here differ from those already described in the literature?

We thank the reviewer for this valuable suggestion. The introduction and discussion sections of the manuscript have been appropriately revised to highlight the study's novel contributions. Please see our above response regarding the study's novelty.

3. I applaud the use of spiked urine samples for technical validation. However, I am concerned about the use of a single strain from the various species. A panel of clinical isolates should be used, at least in the final method validation, to ensure that performance reflects a species average rather than optimization for a particular strain. This may be particularly relevant for organisms like *Proteus mirabilis* and *Pseudomonas aeruginosa*, which can have highly divergent morphotypes (e.g., mucoid variants) that could impact assay performance.

We thank the reviewer for this insightful comment. The species used for technical validation were chosen based on the most common gram-negative and gram-positive uropathogens. We also agree with the reviewer on testing the method's performance across different morphotypes and have conducted additional testing of the optimized methods with a panel of 11 prevalent clinical sequence type strains of uropathogens: *E. coli* (ST131, ST38, ST1193), *E. faecalis* (ST30, ST6), *K. pneumoniae* (ST16, ST147), *P. aeruginosa* (ST111, ST235), and *P. mirabilis* (ST1) (see Supplementary Figures 3 and 4). These were selected based on their frequency in clinical UTI cases and their sequence types within those species. This demonstrates the consistency and robustness of the optimized method across various sequence types of common uropathogens. The results section (Lines 157-167) and the methods section (Lines 597-607) have been revised to include this testing and the corresponding results. Testing across multiple sequence types showed that the optimized method consistently achieved a host depletion of 10^3 folds, underscoring the robustness of the method and supporting the previous conclusions.

4. The authors make cost claims, but do not present sufficient supporting information. A detailed line-item list contributing to cost of goods should be included if such claims are made. Personally, I do not think that cost arguments are necessary for a scientific publication if the methods are robust and address the key problem—rapid and accurate detection of UTI pathogens and their AST profiles. I was left unconvinced by the cost computation and feel that it is largely unnecessary.

We thank the reviewer for their comments. To ensure clarity, a detailed account of the cost calculations has been included in the supplementary data (Supplementary Table 5) of the revised manuscript. Cost is a crucial barrier to implementing methods in clinical settings. This study is the first to provide a comprehensive cost estimation of clinical metagenomics, which may be essential for decision makers and represents an important point of novelty.

5. The discussion section could use better focus. What precisely did the authors find that addresses a gap in the literature? While multiple methods were evaluated and one was optimized, are these results different from what would be expected?

We thank the reviewer for this valuable suggestion. Please see our response regarding the novelty of the study. The discussion section has been properly revised to highlight the study's original contributions. We also clarify the improvements over existing methods as demonstrated by the optimized method.

6. I found the very high number of polymicrobial infections atypical, although it may be understandable given the presumably inpatient complicated UTI cohort. Nonetheless, this point warrants better discussion and the clinical characteristics of your patient cohort should be described.

We agree with the reviewer that cUTI are often associated with polymicrobial infections and thank the reviewer for raising this point. We have clarified the inclusion criteria in the methods section (L615-620): “*The cUTI samples included in the study were obtained from patients exhibiting one or more risk factors such as the presence of a urinary catheter or a previous history of urological intervention.*” The ethical approval limits us from providing any additional

clinical or patient information beyond what has already been shared. The complexity of the sample cohort is an important novelty of the current study.

7. The authors must provide a human ethics protocol number and describe patient recruitment procedures in the Methods section.

Thank you. The methodology section has been revised in the manuscript to provide more detail about the inclusion criteria (Lines 620-624). The ethics protocol number is “AZ 158/20” which was included in the Ethics information section (Lines 944-945) following the references.

8. I was hoping to find more robust benchmarking against established clinical methods. For example, colony counts, microbial identification by MALDI-TOF, and AST profiles by VITEK or broth microdilution could be used as references. Given that these data came from a clinical laboratory, I presume that this information could be made available.

We thank the reviewer for this comment and regret that it was not clear that these data were already included in the manuscript and its supporting information. The manuscript has been revised to better illustrate the benchmarking of metagenomic results against the conventional results obtained from colony counts, MALDI-TOF, and VITEK-2. Additionally, the supplementary information, which includes both metagenomic and conventional data, has been updated and divided into Supplementary Table 2, containing clinical data, and Supplementary Table 3, which presents metagenomic and clinical data, for greater clarity.

9. Similarly, the high rate of polymicrobial infections could be cross-validated with clinical microbiology reference data.

We apologize for the lack of clarity on this point. The polymicrobial infections were identified through routine MALDI-TOF testing as described. The manuscript has been revised to more clearly reflect the benchmarking results. Additional pathogens detected by mNGS were cross-validated with PCR or Vivalytic (Lines 671-679).

10. Portions of the discussion highlight findings that are unsurprising, such as the prevalence of antimicrobial resistance in UTI pathogens, the efficacy of host genomic depletion, and utility arguments related to cost and screening.

The discussion section of the manuscript has been revised to highlight the study's novelty and distinctive contributions.

Figures:

11. Figures 2A and 2B might be better presented as a table or confusion matrix. It is not clear what the expected versus observed outcomes are, which makes it difficult to assess method efficacy.

We thank the reviewer for this valuable suggestion. The figure has been updated in the revised manuscript to include a confusion matrix, which facilitates comparison with the routine culture results. Additionally, we have included a table (Table 1) to provide an overview of all accuracy

metrics for clarity. We have also detailed all calculations in the Methods section (Lines 705-726).

12. Figure 2C lacks a gold standard reference. It is unclear whether species co-occurrence is based on novel metagenomic findings or a traditional culture-based system. This distinction is important.

We thank the reviewer for this point. The figure has been updated in the revised manuscript to clarify benchmarking against traditional culture-based results obtained from MALDI-TOF MS and VITEK 2 compact systems. Co-occurrence plots of pathogen ID and resistance were also based on the same routine findings from MALDI-TOF MS and VITEK 2. The legend in Figure 2 has been revised to provide more explicit clarification.

13. Similar concerns apply to Figures 3A, 3B, and 3C. I would prefer to see a comparison of expected versus observed antibiotic susceptibility results.

We thank the reviewer for this suggestion. The figure has been updated in the revised manuscript to clarify benchmarking against traditional culture-based results obtained from VITEK 2 compact systems. We would like to draw the reviewer's attention to supplementary figures 7 and 8, where the same AST concordance results are stratified in several detailed ways.

14. The ROC curves shown in Figure 4 are clear. However, the concept of DNA screening feels disconnected from the manuscript's main narrative.

We thank the reviewer for this point. Here, we want to emphasize that pre-screening with DNA yield and/or flow cytometry can be used to predict culture positivity. This can lower per-sample costs by identifying which samples should be sequenced and which should not. This adds an important element of novelty to the current study as the cost factor is highly relevant for the potential integration of this method into clinical routine.

Reviewer #3:

1. Line 471: The concentration of human cells/nucleic acids is an important factor affecting the sensitivity of mNGS detection. The authors used WBCs from the human buffy coat in the "spiked urine samples," but did not mention the number or concentration of these cells. This is inappropriate. Please provide this information. What level of human nucleic acid background was used when designing the spiked urine samples and conducting the experiments? In other words, it is unclear whether the "spiked urine samples" had high or low background. A more advanced experimental design would involve preparing quality control samples with varying concentrations of human nucleic acids to evaluate the method's sensitivity under different background conditions. This would better reflect clinical scenarios, where urine samples from patients with UTIs show great variability in leukocyte content. The authors are encouraged to consider this.

We thank the reviewer for raising a very relevant point and agree that the variability of human cells is an important factor influencing the method's sensitivity. However, we chose

to spike whole cells (WBCs) instead of nucleic acids because this approach assesses both host cell lysis and nucleic acid depletion after lysis. Moreover, our ethical approval and concerns about collecting and processing large blood volumes from healthy volunteers prevent us from testing different levels of WBCs because a high host background concentration requires a substantial amount of blood. We also want to respectfully direct the reviewer's attention to Supplementary Figure 12, which demonstrates the wide range of host (10^1 - 10^5 /mL) and bacterial (10^1 - 10^6 /mL) cell counts across various samples, along with the corresponding sequencing reads. This emphasizes the method's sensitivity to different host background levels. The host cell concentrations used in the current spike-in experiments are included in the revised manuscript for each related experiment (Lines 571-573, 587-589, and 597-607).

2. Line 545–546: Please explain in detail the detection range of qPCR and Vivalytic tests.

We thank the reviewer for raising this question. In this section, we used PCR/Vivalytic methods to confirm the additional pathogens detected by mNGS methods. The methodology section has been updated in the revised manuscript to include the detection range.

3. Line 574–575: Please specify the positivity threshold for ARGs. This was overlooked. What parameters did the authors use to define positive versus negative detection?

We thank the reviewer for this point. As described in the methods section, a reference assembly-based annotation method was used for meta-genomic ARG detection. Therefore, no read-based thresholds were applied. Consequently, any ARGs detected in the assembled sequences were included in the analysis. Assemblies with less than 2% reference coverage were excluded, as mentioned in the methods section. This approach shows a high correlation with results from clinical susceptibility testing, as presented in the results.

4. Figures 2c and 3c: These figures are not the most effective ways (figures) to illustrate co-detection of bacteria and resistance genes. Moreover, the figures lack values showing the frequency or proportion of each combination, which is confusing. Consider replacing them with a clearer visualization. Also, for Figure 2 and Supplementary Figure 6, please display the actual values on top of each bar.

We thank the reviewer for this observation. The figures have been updated with a scale to show the number of samples in which the species co-occur (for Figure 2d) and the frequency of resistance to antibiotics (for Figure 3d). The legends have also been adjusted to match. The supplementary figures (11 & 12) have been revised to display the values.

5. Line 105: “patent” should be “patient.”?

Thank you! The wording has been updated in the revised manuscript.

6. Line 186: The authors mentioned the detection of virulence genes here, but the Methods section does not describe how virulence genes were detected or reported. Please add this information.

Thank you for pointing this out. The method section has been revised to include the methodology used for detecting virulence genes (lines 703).

7. Legend of Supplementary Figure 3: "105 CFU/mL" should be written as "10⁵ CFU/mL."

The correction has been made in the revised version of the manuscript.

8. Line 204–205: Why was this statement made? Is there experimental data supporting it, or is it just empirical (which is not appropriate in a scientific paper)? Also, how are “a low pathogen load” and “very high leukocyte load” defined? The current study design does not provide a basis for drawing such conclusions.

We thank the reviewer for this comment. The statement has been updated in the revised manuscript.

9. A general issue throughout the manuscript is that many of the numerical values presented are not supported by raw data or calculation methods. For instance, we do not know how the values in Line 201 (17%), Line 203 (19%), Line 207 (170%), and Line 208 (1%) were derived. Even in Supplementary Figure 7, there are bare plots without any numeric annotations. This appears highly unrigorous.

We appreciate the reviewer's comment. This has been corrected throughout the manuscript. The supporting calculations and raw data have been included in the revised version, along with additional supporting figures and tables.

10. Supplementary Figure 8: Please clearly display both values and percentages above each bar.

We thank the reviewer for the suggestion. Supplementary Figure 12 has been updated to display the values and percentages in the revised manuscript.

11. Line 223–244: Due to various influencing factors during experimentation, DNA yield may show significant inter-batch variation. Furthermore, the complexity of urine varies among patients depending on disease severity. Therefore, using a small sample size to establish a DNA yield threshold for screening is prone to significant error and may not be generalisable to larger or different populations. If the authors intend to demonstrate that this analysis is meaningful or propose DNA yield as a screening or diagnostic marker, they should validate it using an independent cohort. Otherwise, this result should not be highlighted in the main text or abstract.

We thank the reviewer for this suggestion. We agree that various factors can influence DNA yield; therefore, we will further test this metric in our follow-up study, where the method will be evaluated using over 300 clinical urine samples. The discussion section has been revised accordingly.

12. Line 249: It is recommended to present both percentages and absolute values, e.g., “90% (45/50),” to aid understanding.

We thank the reviewer for this suggestion. The necessary corrections have been incorporated throughout the revised manuscript.

13. Line 252: Replace “the selected method” with “the optimized method.”

The corrections have been made in the revised version of the manuscript.

14. Line 252–254: Turnaround time (TAT) usually depends on the number of samples processed per batch. There is no information in the manuscript about how many samples can be tested in one batch. How many samples can be processed on one chip? For clinical application, can the workflow support sample-on-demand testing?

We thank the reviewer for this comment. For the current study, samples were processed in batches of 12 to 24 samples, depending on the number received. The nanopore flow cells currently support multiplexing up to 96 samples per flow cell. If a single on-demand sample test is required, a single sample can be sequenced, and the flow cell can be reused for subsequent samples (up to 96). This demonstrates that nanopore sequencing is a flexible approach that can be tailored to the throughput needs of any lab based on specific requirements (Nanopore, 2025).

15. Line 257: Does the mentioned cost include the reagents used for host depletion?

Thank you for your comment. Yes, the cost per sample mentioned in the manuscript includes the reagents used for host depletion. A supplementary datasheet with detailed cost calculations is included in the manuscript (Supplementary Table 5).

Reviewers' comments:

We thank the reviewers for their valuable comments for improving the quality of the manuscript. Below is the point-by-point response to the comments raised by the reviewers:

Reviewer #1:

1. Figure 1: Protocol details are missing and need to be clearly presented to support the idea that it could be completed within 4 hours.

We appreciate the reviewer's suggestion. This figure is intended to provide a summary and overview of the study methodology. Based on the reviewer's recommendations, we have added some additional information to the figure. However, including too much detail would make the figure cluttered and difficult to interpret. The full details are described in the methodology section. We also added a note to the figure legend to highlight this (Lines 969-971).

- a. For the step where you add 10uL of HL-SAN/M_SAN and incubate at 37C – what is the duration of the incubation?

Added in Figure 1, with additional details as mentioned in lines 546-547 of the manuscript.

- b. How long does the pellet 2x wash step take? Account for resuspension and centrifuge time.

The 2x wash step takes approximately 15 minutes, including two centrifugation steps of 5 minutes each. Please refer to lines 548-549 for the details.

- c. How long was the treatment with the enzymatic lysis buffer for methods 6-9 and how long does the treatment with proteinase K and bead beating take for methods 10 and 11?

The enzymatic treatment for methods 6-9 was done for 30 minutes (Lines 659-660), while the combined bead beating and the proteinase K treatment lasted for 15 minutes (Lines 662-664).

- d. Please provide the exact duration of the “subsequent extraction steps” for methods 6-11.

We thank the reviewer for the comment. We have now added sentences to clarify the time needed for the following extraction steps in the method section. Please refer to the lines 554-556, 660-662, and 667-668.

- e. Please provide a time estimate for the concentration and purification by Ampure XP.

Thank you for your comment. We have detailed this in Figure 5 and Supplementary Figure 15. Briefly, purifying and concentrating the samples with the Ampure XP protocol takes 30 minutes. This duration is included in the overall calculation of the turnaround time.

- f. You are missing a quantification step before starting library preparation. Nanopore suggests doing this by qubit. Please include your quantitation method and the

duration of doing this for 12-24 samples (as this is noted as the most common sample size in your methods)

Thank you for your comment. DNA quantification was performed using the Qubit 4.0 fluorometer with the new high-sensitivity and broad-range kits, which include ready-to-use standard working solutions. This information has been updated in the text. Please refer to lines 688-690. It takes approximately 5 minutes to quantify up to 24 samples.

- g.** The steps of flow cell priming and loading are not included – please include these as well as the time that must be allotted for these steps.

Thank you for your comment. This is already included in the calculated TAT, which we have detailed in Figure 5 and Supplementary Figure 15. The current preparation time for the rapid barcoding kit is 60 minutes, including flow cell priming and loading steps. The attached reference link to the Nanopore resource (<https://nanoporetech.com/document/rapid-sequencing-gdna-barcoding-sqk-rbk114>) offers a brief overview of the time required for each step of library preparation, including flow cell priming and loading.

2. Cost Analysis:

- a.** There seems to be some inaccuracy in the cost analysis presented in lines 328-335. For the sequencing costs, a single minION flow cell is \$800 and the rapid barcoding kit for 24 samples is also \$800. With these numbers, the price per sample would be ~\$67/sample not include any third party reagents and consumables needed for the full procedure.

We thank the reviewer for bringing this to our attention. We purchased 24-pack kits and used flow cells from this pack in the study. Therefore, the cost analysis was based on the price of a MinION 24-pack (<https://store.nanoporetech.com/us/flow-cell-r10-4-1-2025.html>) and the corresponding RBP Library prep kit. This information is detailed in Supplementary Table 5. The cost amounts to \$25 per sample for the flow cell and \$5.60 for all necessary reagents and consumables when multiplexing 24 samples per flow cell, as shown in Supplementary Table 5. We regret that this was not clear in the main text, so we have added these details to the manuscript (Lines 329-337).

- b.** Also I believe there is a typo in line \$335 – it is stated that DNA extraction costs approximately \$6, which I believe should be \$36.

We thank the reviewer and apologize for the lack of clarity. This line refers to the cost per sample when flow cytometry and DNA yield pre-screening can flag a sample as negative, meaning no sequencing is needed. Specifically, it covers only the costs for host depletion and DNA extraction. This cost is approximately \$6 per sample (\$5.80 based on our calculation) for the optimized method, as detailed in the cost analysis in Supplementary Table 5. We have updated the manuscript to clarify this point Lines 329-337.

- c.** Also this cost estimate assumes multiplexing of 24 samples, but wouldn't having to wait for 24 samples to be ready to test affect turnaround time? Even in a high volume practice, it would likely take at least an hour or two for this

many samples to be submitted for testing. This needs to be either factored into the turnaround time, cost, or discussed as a limitation.

We thank the reviewer for this important note regarding sample multiplexing and turnaround time calculation.

(1) Nanopore offers great flexibility in multiplexing because flow cells can be reused. This allows users to sequence any number of samples with different unique barcodes based on availability and then reuse the same flow cell with unused barcodes at no extra cost. We also addressed this point in response to Reviewer 3 in our previous reply: We stated, *“For the current study, samples were processed in batches of 12-24 samples, depending on the number of samples that were received (Lines 652-653). The nanopore flow cells currently support multiplexing up to 96 samples on one flow cell. If sample-on-demand testing is required, a single sample can be sequenced, and the flow cell can be reused for subsequent samples with different barcodes (up to 96). This means nanopore sequencing is a flexible approach that can be adapted to the throughput required by any lab based on the specific needs”*

(2) As the reviewer correctly asserts, the time from taking patient samples to them being received for analysis can vary greatly depending on the volume and the SOPs in an institution or hospital, as well as other factors, and is outside the scope of the reported method. Therefore, the turnaround time and cost analyses are conducted after the samples are received at the lab, where the protocol is initiated. We have added a sentence to clarify this, Lines 325-326. This enables a reader to better assess the differences between methods discussed in the literature. Additionally, this turnaround time calculation aligns with previously published studies. Please find the examples below:

- Zhang et al. 2022 (doi: 10.3389/fmicb.2022.858777) reported a turnaround time of 6 hours. They collected urine samples and froze them so they could be analyzed at the same time. The turnaround time was calculated starting after the samples were thawed, at the beginning of the wet-lab protocol.
- Liu et al. 2023 (doi: 10.1002/ctm2.824) reported a turnaround time of 10 hours, starting at the beginning of the wet-lab protocol.
- Bellankimath et al., 2024 (doi: 10.1007/s10096-024-04929-1) reported a turnaround time of 5 h using spiked samples. The turnaround time was calculated starting at the beginning of the of the wet-lab protocol.

This is not only common practice when reporting rapid metagenomics methods for UTI but also when using different technologies. Below, we highlight how turnaround time is reported for MALDI-TOF and PCR-based methods. Reporting the turnaround time of the presented method, regardless of the institution where it was performed, allows for comparison between different methods.

- Wei et al. 2019 (doi: 10.3389/fmicb.2019.01182) reported a rapid MALDO-TOF based protocol using a lysis-centrifugation-wash protocol to prepare bacterial pellets. The turnaround time calculation starts at the beginning of this wet-lab protocol.

- Brons et al. 2020 (doi: 10.1016/j.mimet.2019.105799) reported a rapid method for detecting uropathogenic *E. coli* using PCR in spiked samples. The turnaround time calculation started at the beginning of the DNA extraction protocol.
3. Given there is no formal analysis of the method to distinguish true case and controls, the claim that the method can “diagnose UTI” cannot be made. UTI diagnosis requires symptoms (i.e. a host immune response) and the presence of causative pathogen. As it stands, it would be more accurate to say this this method can detect bacteriuria in individuals with suspected complicated UTI. The AUROC analysis presented in figure 4 is only assessing the accuracy in discerning culture positive from culture negative. Although the authors have added to the methods section that a control group of samples were collected from patients without symptoms, it is unclear if these all perfectly corresponded to the culture negative samples (or if some were culture positive). There needs to be a true case/control analytical design (ideally of the mNGS prediction) to claim that the methods performs UTI diagnosis. It’s also a bit confusing because in the sentence just before it says “only those samples that tested positive for leukocytes and nitrate were included in the analysis.” Does this include the control samples?

We thank the reviewer for this excellent suggestion. The diagnosis of an UTI is a multifactorial issue and cannot be made by one method alone. There is a recently published consensus that UTI diagnosis requires the presence of local symptoms and signs, an assessment of systemic involvement, a host immune response, assessed by quantified leucocyturia, and the presence of a causative pathogen (Bilsen M et al., Lancet Inf Dis 2024).

As described in the methods section (Lines 625 – 634), the case group in this study (positive samples) all had risk factors, were symptomatic, and tested positive for leukocyte dipstick. These patients were all treated in the urology ward. The negative control cohort had no symptoms, but they had tested positive for leukocyte dipstick. The control cohort was culture-negative. We regret that this was unclear and have rephrased this part of the methods section for clarity (Lines 629-634). Both the case (culture-positive) and control (culture-negative) cohorts were included in the mNGS analysis. The true negative mNGS samples all came from the control cohort. We have also updated the manuscript to clarify this (Lines 173-174).

4. Frankly, the authors response to my question about CFU/mL estimation does not makes sense. Colony forming unit assays are the gold standard for determining the absolute abundance of a given bacterial pathogen in a clinical sample. It is NOT measured after incubation. The urine is plated directly onto the agar and each colony that arises is representative of a single bacterium that founded it. It is exactly an estimation of the number of bacteria in the sample at the time of plating. The issue with current clinical urine culture methods is not that CFU/mL counts of pathogens are not accurate, its that they can fail to capture more fastidious pathogens that need specialized conditions to grow. You state correctly that mNGS alone only gives relative abundance, so if it needs to be coupled with flow cytometry to gain an accurate estimate of absolute abundance, the time for this additional method should be added to the overall turnaround time if the goal is more than just pathogen detection.

We thank the reviewer and regret that this was unclear in our previous response. We agree with the reviewer and would like to clarify that the urine was not incubated before plating. It is standard practice to plate urine on specific agar plates for CFU calculations immediately after the sample reaches the lab. These plates must then be incubated for 24 hours, after which the CFUs are counted and used to calculate bacteria per mL of urine at the time of plating. Each CFU generally represents an individual bacterium in the urine.

This is correct. In all urology patient samples in the current study, flow cytometry is performed in parallel with routine pathogen identification. Therefore, in this study, flow cytometry was also performed in parallel with the mNGS analysis. This step is fully automated, taking less than an hour, and since it is run in parallel, there is no additional analysis time. We have updated the discussion section to clarify the point (lines 487-489)

5. To clearly show benchmarking of antimicrobial susceptibility in a clinically useful way, the authors need to include Supplemental Figure 8 in the main text. Showing the performance of mNGS for AST prediction by antibiotic and pathogen is exactly the point of the analysis you are showing and is more useful to the reader than Figure 3d. Co-occurrence of antibiotic resistance in samples is not the main point of this analysis and this sub-figure should be moved to the supplement. The figures in Supp Fig 8 should replace it in the main text figure 3.

We thank the reviewer for the relevant suggestion. Figure 3.d and the supplementary figure 8 have been modified accordingly to reflect the same.

6. The title should be changed to more accurately represent the findings of the study based on my comments above that may lead to a revision of the actual TAT and a needed clarification that the mNGS method alone has been shown to perform pathogen and/or bacteriuria detection but not formally UTI diagnosis.

We thank the reviewer for the comment and the thorough review of our manuscript.

We agree with the reviewer and have pointed out above that the diagnosis of a UTI is a multifaceted issue, comprising the assessment of local symptoms and signs, evaluation of systemic involvement, assessment of the host immune response, and identification of the causative pathogen and its antibiotic susceptibility.

We have revised the manuscript title and conclusion, replacing the word 'diagnosis' with 'pathogen identification and antibiotic susceptibility profiling.'

Additionally, we have added more details about the turnaround time in response to the reviewers' comments. The total turnaround time for the optimized method was calculated to be 222 to 245 minutes (see table below). For simplicity, we rounded this to four hours in the manuscript. We believe the revised manuscript now clearly supports the claimed turnaround time (see our responses above). Please refer to the table below for a detailed breakdown of the different steps and their turnaround times in minutes.

Host depletion	DNA extraction	Ampure XP	Library preparation	Sequencing & data analysis
70	60	30	60	2 – 25

Reviewer #1 (Remarks to the Author):

1. I think four hours is a very optimistic estimate and do not fully believe that both pathogen ID and antimicrobial gene profiling will reproducibly be achieved in this time frame. I suggest the addition of "approximately" to the title and other sentences where the four hour time window is discussed.

We agree. The title and the manuscript have been modified to include approximately.

2. Antimicrobial resistance gene identification is not the same as antimicrobial susceptibility testing. The use of "antibiotic susceptibility" in the title makes the reader believe that AST can be done by this method when what is actually being done is AMR gene profiling. Even if the predictions of from the AMR gene profiling are relatively concordant with AST, "antibiotic susceptibility profiling" should not be used to describe the detection of AMR genes due to its connotation. I suggest replacing "susceptibility" with "gene" in the title to avoid confusion.

Thank you for the suggestion. We want to clarify that our study not only involved detecting AMR genes but also identifying mutations related to known resistance mechanisms. We also extensively mapped and correlated these AMR profiles with clinical microbiology antibiotic susceptibility data obtained using Vitek 2. For instance, we explained how low expression levels of the ampC gene in E. coli result from mutations in the promoter and attenuator regions, rather than being induced by antibiotic exposure, such as β -lactams. Furthermore, we conducted a comprehensive analysis, even for antibiotics not routinely tested. Although correlating ARG data with phenotypic resistance can be challenging, our results, with 90% accuracy, demonstrate the potential to extract valuable antibiotic susceptibility information from complex mixed samples.